# FEDERATED SEMI-SUPERVISED LEARNING WITH DUAL REGULATOR

## ABSTRACT

Federated learning emerges as a powerful method to learn from decentralized heterogeneous data while protecting data privacy. Federated semi-supervised learning (FSSL) is even more practical and challenging, where only a fraction of data can be labeled due to high annotation costs. Existing FSSL methods, however, assume independent and identically distributed (IID) labeled data across clients and consistent class distribution between labeled and unlabeled data within a client. In this work, we propose a novel FSSL framework with dual regulator, FedDure, to optimize and customize model training according to specific data distributions of clients. FedDure lifts the previous assumption with a coarse-grained regulator (C-reg) and a fine-grained regulator (F-reg): C-reg regularizes the updating of the local model by tracking the learning effect on labeled data distribution; F-reg learns an adaptive weighting scheme tailored for unlabeled instances in each client. We further formulate the client model training as bi-level optimization that adaptively optimize the model in the client with two regulators. Theoretically, we show the convergence guarantee of the dual regulator. Empirically, we demonstrate that FedDure is superior to the existing methods across a wide range of settings, notably by more than 12% on CIFAR-10 and CINIC-10 datasets.

## 1 INTRODUCTION

Federated learning (FL) is an emerging privacy-preserving machine-learning technique (McMahan et al., 2017), where multiple clients collaboratively learn a model under the coordination of a central server without exchanging private data. Edge devices like mobile phones have generated and stored a large amount of private data. Centralizing these data could lead to data privacy issues (Voigt & Von dem Bussche, 2017). Federated learning is a decentralized learning paradigm to leverage these data and has empowered a wide range of applications in many industries, including healthcare (Kaissis et al., 2020; Li et al., 2019), consumer products (Hard et al., 2018; Niu et al., 2020), and public security (Zhuang et al., 2022).

The majority of existing FL works (McMahan et al., 2017; Wang et al., 2020; Li et al., 2021a) assume that the private data in clients are fully labeled, but the assumption is unrealistic in real-world federated applications as annotating data is time-consuming, laborious, and expensive. To remedy these issues, federated semi-supervised learning (FSSL) is proposed to improve model performance with a large amount of unlabeled data on each client. In particular, priors work (Jeong et al., 2021) has achieved competitive performance by exploring the inter-client mutual knowledge, *i.e.*, inter-client consistency loss Jeong et al. (2021). However, they usually focus on mitigating inter-client heterogeneous data distribution across clients (**External Imbalance**) and assume that the class distribution between the labeled and unlabeled data is consistent. These assumptions enforce strict requirements of data annotation and would not be practical in many real-world applications. A general case is that labeled data and unlabeled data have different data distribution (**Internal Imbalance**). For example, photo gallery in mobile phones contains much more unlabeled images and irrelevant samples than the ones that can be labeled manually for image classification task Yang et al. (2011).

Besides, these existing methods require sharing of additional information among clients, which could impose potential privacy risks. Specifically, they transmit models among clients to provide consistency regularization. However, inter-client interactions might open a loophole to unauthorized infringement for privacy risks Chen et al. (2019); many reverse-engineering techniques Yin et al.

(2020) can even recover the client data from the mutual models, threatening the users' data privacy and security (Yonetani et al., 2017; Wu et al., 2018).

In this paper, we propose a flexible federated semi-supervised learning framework with dual regulator, termed **FedDure**, to handle the internal and external imbalance problems while accounting for privacy. FedDure explores the adaptive regulators to flexibly update the model parameters in clients; it dynamically adjusts dual regulators to optimize the model training in each client according to the learning process and outcome of the client's specific data distribution. Our framework includes two novel components: Coarse-grained regulator (**C-reg**) and Fine-grained regulator (**F-reg**). On the one hand, C-reg regularizes the updating of the local model by tracking the learning effect on labeled data distribution. It mitigates the distribution mismatching between labeled and unlabeled data to prevent corrupted pseudo labels and maintain generalization ability. On the other hand, F-reg learns an adaptive weighting scheme tailored for each client; it automatically equips a *soft* weight for each unlabeled instance to measure its contribution to the training. This scheme automatically adjusts the instance-level weights to strengthen (or weaken) its confidence according to the feedback of F-reg on the labeled data. FedDure utilizes the bi-level optimization strategy to alternately update the local model and dual regulators in clients. We theoretically show that C-reg and F-reg converge with guarantee and empirically demonstrate that FedDure outperforms other methods on various benchmarks.

The main contributions of this work are three-fold. (1) We address a more practical scenario of FSSL, where data distribution is different not only across clients (external imbalance) but also between labeled and unlabeled data within a client (internal imbalance). (2) We propose a flexible FSSL framework with dual regulator, **(FedDure)**, which designs adaptive regulators to flexibly update the local model according to the learning processes and outcomes on specific data distributions of each client. It does not require sharing of additional information among clients. (3) We theoretically analyze and present the convergence of the dual regulator and empirically demonstrate that FedDure is superior to the state-of-the-art FSSL approaches across multiple benchmarks.

## 2 RELATED WORK

**Federated Learning (FL)** is an emerging distributed training technique that trains models on decentralized clients and aggregates model updates in a central server (Yang et al., 2019). It protects data privacy as raw data are always kept locally. FedAvg (McMahan et al., 2017) is a pioneering work that aggregates model updates by weighted averaging. Statistical heterogeneity is an important challenge of FL in real-world scenarios, where the data distribution is inconsistent among clients (Li et al., 2020a). A plethora of works have been proposed to address this challenge with approaches like extra data sharing, regularization, new aggregation mechanisms, and personalization (Zhao et al., 2018; Li et al., 2020b; Zhuang et al., 2020; Li et al., 2021b; Gao et al., 2022). However, these approaches commonly consider only supervised learning settings and may not be simply applied to scenarios where only a small portion of data is labeled. Numerous studies focus on purely unsupervised federated learning, but they are either application-specific or only learn generic visual representations (Zhuang et al., 2021a;b); they do not effectively leverage the small fraction of labeled data that could exist in real-world applications. Our work primarily focuses on federated semi-supervised learning, where a small fraction of data has labels in each client.

**Semi-Supervised Learning** aims to utilize unlabeled data for performance improvements and is usually divided into two popular branches pseudo labeling and consistency regularization. Pseudo-labeling methods (Lee et al., 2013; Zou et al., 2022; Pham et al., 2021) usually generate artificial labels of unlabeled data from the model trained by labeled data and apply the filtered high-confidence labels as supervised signals for unlabeled data training. MPL Pham et al. (2021) extends the knowledge distillation and meta-learning to SSL by optimizing the teacher model with feedback from the student model. Consistency regularization (Lee et al., 2022; Tarvainen & Valpola, 2017) regularizes the outputs of different perturbed versions of the same input to be consistent. Many works (Sohn et al., 2020; Zhang et al., 2021a; Lee et al., 2022) apply data augmentation as a perturbed strategy for pursuing outcome consistency. FixMatch Sohn et al. (2020) follows the UDA and brings the idea of pseudo-label to model training with unlabeled samples filtered by fixed threshold.

**Federated Semi-Supervised Learning (FSSL)** considers learning models from decentralized clients where a small amount of labeled data resides on either clients or the server Jin et al. (2020).

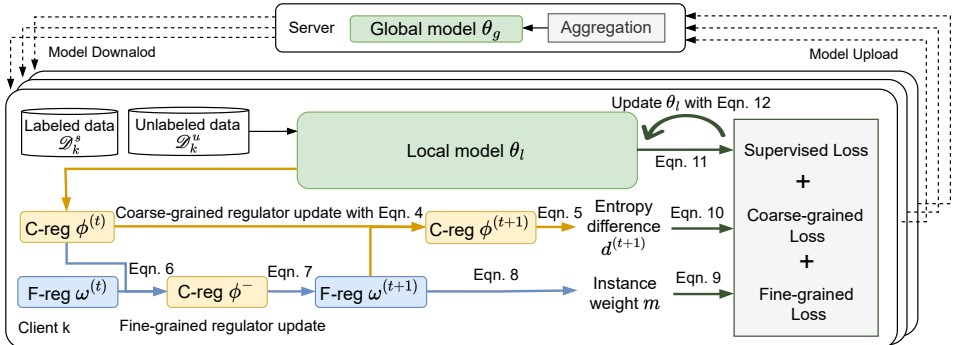

Figure 1: **Illustration of Federated Semi-Supervised Learning Framework with Dual Regulator (FedDure)**. FedDure updates the coarse-grained regulator (C-reg) and the fine-grained regulator (F-reg) to adaptively regularize the model training in each client: C-reg dynamically regulates the importance of local training on the unlabeled data by reflecting the overall learning effect on labeled data; F-reg regulates the performance contribution of each unlabeled sample.

FSSL scenarios can be classified into three categories: (1) Labels-at-Server assumes that clients have purely unlabeled data and the server contains some labeled data (Lin et al., 2021; He et al., 2021; Zhang et al., 2021b; Diao et al., 2021); (2) Labels-at-Clients considers each client has mostly unlabeled data and a small amount of labeled data (Jeong et al., 2021); (3) Labels-at-Partial-Clients assumes that the majority of clients contain fully unlabeled data while numerous clients have fully labeled data (Zhang et al., 2021b; Lin et al., 2021; Liang et al., 2022). Labels-at-Clients has been largely overlooked; the prior work (Jeong et al., 2021) proposes inter-client consistency loss, but it shares extra information among clients and bypasses the internal class imbalance issue. This work introduces the dual regulator to address the issue, with no extra information shared among clients.

**Class Imbalance Methods** are concerned with dataset resampling (Chawla et al., 2002; Buda et al., 2018) or loss reweighting (Ling & Sheng, 2008; Sun et al., 2007; Cui et al., 2019) for gradient calculation. In the centralized learning setting, many methods (Chawla et al., 2002; Liu et al., 2008) focus on resampling from the minority class for balanced class-wise distribution. Important examples receive more attention and align larger weights than others for accelerating the optimization of networks. The important examples are quantified by their loss Zhao & Zhang (2015) or the uncertainty Chang et al. (2017). Besides, the loss reweighting methods usually formulate the sampling scheme on certain prior knowledge. Typical methods include the AdaBoost Freund & Schapire (1997) and focal loss Lin et al. (2017), which focus on those hard examples and align larger weights for them.

## 3 METHOD

This section first defines the problem and introduces a novel framework with dual regulators (Fed-Dure). Using the dual regulator, we then build a bi-level optimization strategy for federated semi-supervised learning.

### 3.1 PROBLEM DEFINITION

We focus on Federated Semi-Supervised Learning (FSSL) with external and internal class imbalance problems, where each client data is partially labeled. We assume that there are $K$ clients, denoted as $\{\mathcal{C}_1, ..., \mathcal{C}_K\}$. Federated learning aims to train a generalized global model $f_g$ with parameter $\theta_g$. It coordinates decentralized clients to train their local models $\mathcal{F}_l = \{f_{l,1}, ..., f_{l,K}\}$ with parameters $\{\theta_{l,1}, ..., \theta_{l,K}\}$, where each client is only allowed to access its own local private dataset. In the standard semi-supervised setting, the dataset contains a labeled set $\mathcal{D}^s = \{\mathbf{x}_i, \mathbf{y}_i\}_{i=1}^{N^s}$ and an unlabeled set $\mathcal{D}^u = \{\mathbf{u}_i\}_{i=1}^{N^u}$, where $N^s \ll N^u$. Under FSSL, the private dataset $\mathcal{D}_k$ of each client $C_k$ contains $N_k^s$ labeled instances $\mathcal{D}_k^s = \{\mathbf{x}_{i,k}, \mathbf{y}_{i,k}\}_{i=1}^{N_k^s}$ and $N_k^u$ unlabeled instances $\mathcal{D}_k^u = \{\mathbf{u}_{i,k}\}_{i=1}^{N_k^u}$.

In this work, we primarily focus on image datasets. For an unlabeled image $\mathbf{u}_k$ in client $C_k$, we compute the corresponding pseudo label $\hat{\mathbf{y}}_k$ with the following equation:

$$\hat{\mathbf{y}}_k = \text{argmax}(f_{l,k}(\mathcal{T}_w(\mathbf{u}_k); \boldsymbol{\theta}_{l,k})), \tag{1}$$

where $\mathcal{T}_w(\mathbf{u}_k)$ is the weakly-augmented version of $\mathbf{u}_k$ and the pseudo labeling dataset in the client $C_k$ is denoted as $\mathcal{D}_k^u = \{\mathbf{u}_{i,k}, \hat{\mathbf{y}}_{i,k}\}_{i=1}^{N_k^u}$. For simplicity of notation, we omit the symbol $k$ in the parameters later.

## 3.2 DUAL REGULATOR

In this section, we present federated semi-supervised learning with dual regulator, termed FedDure. FedDure adaptively adjusts gradient updates in each client according to the class distribution characteristics with two regulators, a coarse-grained regulator (C-reg) and a fine-grained regulator (F-reg). Figure 1 depicts the optimization process with these two regulators. We first introduce the regulators in this section and present the optimization process in Section 3.3.

**Coarse-grained Regulator (C-reg).** C-reg dynamically regulates the importance of local training on the unlabeled data by quantifying the learning effect using labeled data. We define C-reg as $f_d$ with parameters $\boldsymbol{\phi}$. Intuitively, C-reg and local models are collaboratively optimized during the local training process in each client. On the one hand, C-reg is trained using pseudo labels generated by the local model in a supervised manner. On the other hand, the parameters of the local model are further rectified according to the feedback from C-reg. Specifically, at training iteration $t$, the coarse-grained regulator searches its optimal parameter $\boldsymbol{\phi}^*$ by minimizing the cross-entropy loss on unlabeled data with pseudo labels. We formulate it as:

$$\boldsymbol{\phi}^* = \underset{\boldsymbol{\phi}}{\text{argmin}}\, \mathbb{E}_{\mathbf{u}}\left[\mathcal{L}_{ce}\left(\hat{\mathbf{y}}, f_d(\mathcal{T}_s(\mathbf{u}); \boldsymbol{\phi}^{(t)})\right)\right], \quad w.r.t \quad \hat{\mathbf{y}} = \text{argmax}(f_l(\mathcal{T}_w(\mathbf{u}); \boldsymbol{\theta}_l)), \tag{2}$$

where $\mathcal{T}_s(\mathbf{u})$ is the strongly-augmented unlabeled image $\mathbf{u}$. The optimal parameter $\boldsymbol{\phi}^*$ is related to the local model's parameter $\boldsymbol{\theta}_l$ via the generated pseudo label, where we denote the relationship as $\boldsymbol{\phi}^*(\boldsymbol{\theta}_l)$. In practice, it requires heavy computational costs to explore the optimal parameter by training the coarse-grained regulator, so we design a strategy to approximate $\boldsymbol{\phi}^*$.

Conceptually, we approximate $\boldsymbol{\phi}^*$ by performing *one gradient step* based on the parameter at training iteration $t$ (i.e., $\boldsymbol{\phi}^{(t)}$). We establish the meta learning process between $\boldsymbol{\phi}$ and $\boldsymbol{\theta}$ as followed:

$$\boldsymbol{\phi}^*(\boldsymbol{\theta}_l^{(t)}) \approx \boldsymbol{\phi}^{(t+1)}(\boldsymbol{\theta}_l^{(t)}) = \boldsymbol{\phi}^{(t)} - \eta_s \nabla_{\boldsymbol{\phi}^{(t)}} \mathbb{E}_{\mathbf{u}} \mathcal{L}_{ce}\left(\hat{\mathbf{y}}, f_d\left(\mathcal{T}_s(\mathbf{u}); \boldsymbol{\theta}_l^{(t)}; \boldsymbol{\phi}^{(t)}\right)\right). \tag{3}$$

Practically, we update C-reg by utilizing the updated fine-grained regulator to measure the adaptive weight for each unlabeled instance, where the updated fine-grained regulator is obtained based on *one gradient step* of C-reg ($\boldsymbol{\phi}^-$ in Eqn. 6). The formulation to optimize C-reg is as followed:

$$\boldsymbol{\phi}^{(t+1)} = \boldsymbol{\phi}^{(t)} - \eta_s \mathcal{H}(\boldsymbol{w}^{(t+1)}; \boldsymbol{\phi}^{(t)}) \nabla_{\boldsymbol{\phi}} \mathbb{E}_{\mathbf{u}}\, \mathcal{L}_{ce}\left(\hat{\mathbf{y}}, f_d\left(\mathcal{T}_s(\mathbf{u}); \boldsymbol{\phi}^{(t)}\right)\right)\Big|_{\boldsymbol{\phi}^{(t)}}, \tag{4}$$

where $\mathcal{H}(\boldsymbol{w}^{(t+1)}; \boldsymbol{\phi}^{(t)}) = f_w\left(f_d\left(\mathcal{T}_s(\mathbf{u}); \boldsymbol{\phi}^{(t)}\right); \boldsymbol{w}^{(t+1)}\right)$, $f_w$ is the fine-grained regulator (F-reg), and $\boldsymbol{w}^{(t+1)}$ is the parameters of F-reg updated by Eqn. 7 that we present in the following subsection.

Next, we quantify the learning effect of the local model with the C-reg using labeled samples by computing the entropy difference $d^{(t+1)}$ of C-reg between training iterations $t$ and $t + 1$:

$$d^{(t+1)} = \mathbb{E}_{\mathbf{x},\mathbf{y}}\left[\left(\mathcal{L}_{ce}\left(\mathbf{y}, f_d(\mathbf{x}; \boldsymbol{\phi}^{(t)})\right) - \mathcal{L}_{ce}\left(\mathbf{y}, f_d(\mathbf{x}; \boldsymbol{\phi}^{(t+1)})\right)\right)\right]. \tag{5}$$

The learning effect is further used as reward information to optimize the local model by regulating the importance of local training on unlabeled data. In particular, the cross-entropy differences signify the generalization gap for the C-reg updated by the pseudo labels from the local model. We provide detailed derivation in Appendix A.4 for the optimization of the local model with C-reg.

**Fine-grained Regulator (F-reg).** F-reg regulates the importance of each unlabeled instance in local training. It addresses the challenge that the $f_d$ is substantially hindered by corrupted labels or class imbalance, especially in the early period of training rounds. This challenge could further negatively impact the optimization of local model $f_l$. Furthermore, previous methods usually utilize

a *fixed threshold* to filter noisy pseudo labels in all clients. It is infeasible for clients with heterogeneous data when the skewed data distribution across clients is unknown.

To tackle the challenge, we construct F-reg $f_w$ parameterized by $\boldsymbol{w}$. [1] It learns an adaptive weighting scheme tailored for each client according to the unlabeled data distribution characteristics. A unique weight is generated for each unlabeled image to measure the contribution of the image to the overall performance. We first perform *one gradient step* update of C-reg $\boldsymbol{\phi}$ to associate F-reg and C-reg with the following formula:

$$\boldsymbol{\phi}^-(\boldsymbol{w}^{(t)}) = \boldsymbol{\phi}^{(t)} - \eta_s \mathcal{H}(\boldsymbol{w}^{(t)}; \boldsymbol{\phi}^{(t)}) \nabla_{\boldsymbol{\phi}} \mathbb{E}_{\boldsymbol{u}} \, \mathcal{L}_{ce} \left( \hat{\mathbf{y}}, f_d \left( \mathcal{T}_s(\mathbf{u}); \boldsymbol{\phi}^{(t)} \right) \right) \Big|_{\boldsymbol{\phi}^{(t)}}, \tag{6}$$

where *one gradient step* of C-reg $\phi^-$ depends on the F-reg $\boldsymbol{w}^{(t)}$. Next, we optimize F-reg in local training iteration $t$ as follows:

$$\boldsymbol{w}^* \approx \boldsymbol{w}^{(t+1)} = \boldsymbol{w}^{(t)} - \eta_w \nabla_{\boldsymbol{w}^{(t)}} \mathbb{E}_{\mathbf{x},\mathbf{y}} \mathcal{L}_{ce} \left( \mathbf{y}, f_d(\mathbf{x}; \boldsymbol{\phi}^-(\boldsymbol{w}^{(t)})) \right), \tag{7}$$

where $f_d(\mathbf{x}; \boldsymbol{\phi}^-(\boldsymbol{w}^{(t)}))$ is the output of $f_d$ on labeled data. We then introduce a re-weighting scheme that calculates a unique weight $\boldsymbol{m}_i$ for $i$-th unlabeled sample:

$$\boldsymbol{m}_i = f_w(f_l(\mathcal{T}_s(\mathbf{u}_i), \boldsymbol{\theta}_l^{(t)}), \boldsymbol{w}^{(t+1)}). \tag{8}$$

Note that $m_i$ is a scalar to re-weight the importance of the corresponding unlabeled image.

## 3.3 BI-LEVEL OPTIMIZATION

In this section, we present optimization processes for the dual regulator and local model $\theta$. We alternatively train two regulators, which approximate a gradient-based bi-level optimization procedure (Finn et al., 2017; Liu et al., 2018). Then, we update the local model with fixed C-reg and F-reg.

**Update F-reg.** Firstly, we obtain one gradient step update of C-reg $\phi^-$ using Eqn. 6. After that, the supervised loss $\mathcal{L}_{ce} \left( \mathbf{y}, f_d(\mathbf{x}; \boldsymbol{\phi}^{(-)}(\boldsymbol{w}^{(t)})) \right)$ guides the update of the F-reg with Eqn. 7.

**Update C-reg.** After updating the parameters of F-reg, we update C-reg by Eqn. 4, regarding local model $\boldsymbol{\theta_l}^{(t)}$ and the updated F-reg $\boldsymbol{w}^{(t+1)}$ as fixed parameters.

**Update Local Model with F-reg.** We use the updated F-reg $\boldsymbol{w}^{(t+1)}$ to calculate a unique weight $\boldsymbol{m}_i$ for $i$-th unlabeled sample with Eqn. 8. The gradient optimization is formulated as:

$$\boldsymbol{g}_u^{(t)} = \mathbb{E}_{\mathbf{u}} \left[ \boldsymbol{m} \cdot \nabla_{\boldsymbol{\theta}_l^{(t)}} \mathcal{L}_{ce} \left( \hat{\mathbf{y}}, f_l \left( \mathcal{T}_s(\mathbf{u}); \boldsymbol{\theta}_l^{(t)} \right) \right) \right]. \tag{9}$$

**Update Local Model with C-reg.** We then use C-reg to calculate entropy difference $d^{(t+1)}$ using Eqn. 5. The entropy difference $d^{(t+1)}$ is adopted as a reward coefficient to adjust the gradient update of local model on unlabeled data. The formulation is as followed:

$$\boldsymbol{g}_d^{(t)} = d^{(t+1)} \cdot \nabla_{\boldsymbol{\theta}_l^{(t)}} \mathbb{E}_{\mathbf{u}} \mathcal{L}_{ce} \left( \hat{\mathbf{y}}, f_l \left( \mathcal{T}_s(\mathbf{u}); \boldsymbol{\theta}_l^{(t)} \right) \right), \tag{10}$$

where the learning process can be derived by meta-learning strategy shown in Appendix A.4.

**Update Local Model with Supervised Loss.** Besides, we compute the gradient local model on labeled data as followed:

$$\boldsymbol{g}_s^{(t)} = \nabla_{\boldsymbol{\theta}_l^{(t)}} \mathbb{E}_{\mathbf{x},\mathbf{y}} \mathcal{L}_{ce} \left( \mathbf{y}, f_l \left( \mathbf{x}; \boldsymbol{\theta}_l^{(t)} \right) \right). \tag{11}$$

On this basis, we update the local model's parameter with the above gradient computation in Eqn. 9, 10 and 11, which is defined as:

$$\boldsymbol{\theta}_l^{(t+1)} = \boldsymbol{\theta}_l^{(t)} - \eta \left( \boldsymbol{g}_s^{(t)} + \boldsymbol{g}_u^{(t)} + \boldsymbol{g}_d^{(t)} \right), \tag{12}$$

where $\eta$ denotes the learning rate of the local model. Finally, after $T$ local epochs, the local model is returned to the central server. The server updates the global model $\boldsymbol{\theta_g}^{r+1}$ by weighted averaging the parameters from these received local models in the current round, and the $r+1$ round is conducted by sending $\boldsymbol{\theta_g}^{r+1}$ to the randomly selected clients as initialization. Alg. 1 presents the pipeline of the overall optimization process.

---

[1]F-reg is a MLP architecture with one fully connected layer with 128 filters and a Sigmoid function.

---

**Algorithm 1** Federated Semi-supervised Learning with Dual Regulator (FedDure)

---

**Require:** $K$: number of clients; $S$: number of selected clients in each round; $R$: number of training rounds; $T$: number of local iterations;

1: **RunServer**
2: Initialize $\theta_g$ and $w$ for each client
3: **for** each round $r$ from 0 to $R-1$ **do**
4:     Randomly select $\{\mathcal{C}_k\}_{k=1}^S$ from $K$ clients;
5:     **for** each $k \in [1, S]$ **in parallel do**
6:         $\theta_{l,k}^{(r+1)} \leftarrow$ **RunClient**$(\theta_g^r)$
7:     **end for**
8:     $\theta_g^{(r+1)} \leftarrow \frac{1}{\sum_{k=1}^S (|\mathcal{D}_k^s| + |\mathcal{D}_k^u|)} \sum_{k=1}^S (|\mathcal{D}_k^s| + |\mathcal{D}_k^u|) \cdot \theta_{l,k}^{(r+1)}$           ▷ Aggregation
9: **end for**
10: **return** $\theta_g^R$;
11: **RunClient**$(\theta_g^r)$
12: $\theta_l^{(0)} \leftarrow \theta_g^r; \phi^{(0)} \leftarrow \theta_g^r$
13: **for** each local iteration $t$ from 0 to $T-1$ **do**
14:     **for** minibatch $\widetilde{\mathcal{D}}_k^u \in \mathcal{D}_k^u$ and $\widetilde{\mathcal{D}}_k^s \in \mathcal{D}_k^s$ **do**
15:         $\hat{\mathbf{y}}_{i,k} \leftarrow$ Generate pseudo labels for unlabeled data $\widetilde{\mathcal{D}}_k^u$ with Eqn. 1; $\widetilde{\mathcal{D}}_k^u \leftarrow \{\mathbf{u}_{i,k}, \hat{\mathbf{y}}_{i,k}\}_{i=1}^{N_k^u}$
16:         Update the fine-grained regulator $w$ with Eqn. 7
17:         Compute the instance weight $m_i$ with Eqn. 8
18:         Update the coarse-grained regulator $\phi$ with Eqn. 4
19:         Compute the entropy difference $d$ with Eqn. 5
20:         Compute local model's gradient $\boldsymbol{g}_u^{(t)}, \boldsymbol{g}_d^{(t)}, \boldsymbol{g}_s^{(t)}$ following Eqn. 9, 10, and 11
21:         Update local model $\theta_l$ with Eqn. 12
22:     **end for**
23: **end for**
24: **return** $\theta_l^T$

---

## 3.4 Convergence of Optimization Process

In this section, we further analyze the convergence of the coarse-grained and the fine-grained regulators and derive the following theorems. The proofs are provided in Appendix A.5.

**Theorem 1** *Suppose supervised loss $\mathcal{L}_{ce}(\boldsymbol{y}, f_d(\boldsymbol{x}; \boldsymbol{\phi}^{(t+1)}(\boldsymbol{\theta}_l^{(t)})))$ is L-Lipschitz and has $\rho$-bounded gradients. The $\mathcal{L}_{ce}\left(\hat{\boldsymbol{y}}, f_d\left(\mathcal{T}_s(\boldsymbol{u}); \boldsymbol{\phi}^{(t)}\right)\right)$ has $\rho$-bounded gradients and twice differential with Hessian bounded by $\mathcal{B}$. Let the learning rate $\eta_s = \min\{1, \frac{e}{T}\}$ for constant $e > 0$, and $\eta = \min\{\frac{1}{L}, \frac{c}{\sqrt{T}}\}$ for some $c > 0$, such that $\frac{\sqrt{T}}{c} \geq L$. Thus, the optimization of the local model using coarse-grained regulator can achieve:*

$$\min_{0 \leq t \leq T} \mathbb{E}[\|\nabla_{\theta_l} \mathcal{L}_{ce}(\boldsymbol{y}, f_d(\boldsymbol{x}; \boldsymbol{\phi}^{(t+1)}(\boldsymbol{\theta}_l^{(t)})))\|_2^2] \leq \mathcal{O}(\frac{c}{\sqrt{T}}). \tag{13}$$

**Theorem 2** *Suppose supervised and unsupervised loss functions are Lipschitz-smooth with constant $L$, and have $\rho$-bounded gradient. The $\mathcal{H}(\cdot)$ is differential with a $\epsilon$-bounded gradient and twice differential with its Hessian bounded by $\mathcal{B}$. Let learning rate $\eta_s$ satisfies $\eta_s = \min\{1, \frac{k}{T}\}$ for constant $k > 0$, such that $\frac{k}{T} < 1$. $\eta_w = \min\{\frac{1}{L}, \frac{c}{\sqrt{T}}\}$ for constant $c > 0$ such that $\frac{\sqrt{T}}{c} \geq L$. The optimization of the coarse-grained regulator can achieve:*

$$\lim_{t \to \infty} \mathbb{E}[\|\mathcal{H}(\boldsymbol{w}^{(t+1)}; \boldsymbol{\phi}^{(t)}) \nabla_\phi \mathbb{E}_{\boldsymbol{u}} \, \mathcal{L}_{ce}\left(\hat{\boldsymbol{y}}, f_d\left(\mathcal{T}_s(\boldsymbol{u}); \boldsymbol{\phi}^{(t)}\right)\right)\Big|_{\boldsymbol{\phi}^{(t)}}\|_2^2] = 0. \tag{14}$$

## 4 Experiments

In this section, we demonstrate the effectiveness and robustness of our method through comprehensive experiments in three benchmark datasets with multiple data settings. More details and additional experiments can be found in the supplementary material.

Table 1: Performance comparison of our proposed FedDure with state-of-the-art methods on three different data heterogeneity settings. FedDure outperforms all other methods in all settings.

| Methods | CIFAR10 | | | Fashion-MNIST | | | CINIC-10 | | |
|---|---|---|---|---|---|---|---|---|---|
| | (IID, IID) | (IID, DIR) | (DIR, DIR) | (IID, IID) | (IID, DIR) | (DIR, DIR) | (IID, IID) | (IID, DIR) | (DIR, DIR) |
| FedAvg* | $45.68 \pm 1.14$ | $43.83 \pm 0.36$ | $40.34 \pm 0.46$ | $85.56 \pm 0.09$ | $84.84 \pm 0.09$ | $82.24 \pm 0.07$ | $40.73 \pm 0.50$ | $39.00 \pm 0.31$ | $28.09 \pm 0.53$ |
| FedAvg-SL | $75.47 \pm 0.41$ | $66.70 \pm 0.86$ | $58.38 \pm 0.41$ | $89.87 \pm 0.23$ | $88.60 \pm 0.25$ | $86.95 \pm 1.12$ | $67.97 \pm 0.50$ | $57.72 \pm 1.67$ | $46.21 \pm 1.06$ |
| FedProx-SL | $74.67 \pm 0.55$ | $66.78 \pm 0.87$ | $59.55 \pm 0.61$ | $89.53 \pm 0.23$ | $88.35 \pm 0.02$ | $87.32 \pm 0.84$ | $68.13 \pm 0.96$ | $58.67 \pm 1.04$ | $52.09 \pm 0.20$ |
| FedAvg+UDA | $47.47 \pm 0.67$ | $43.89 \pm 0.15$ | $35.52 \pm 0.52$ | $86.20 \pm 0.75$ | $85.35 \pm 0.62$ | $81.07 \pm 0.56$ | $42.25 \pm 0.31$ | $39.93 \pm 0.57$ | $29/27 \pm 0.09$ |
| FedProx+UDA | $46.49 \pm 0.74$ | $42.82 \pm 0.79$ | $37.38 \pm 0.52$ | $84.78 \pm 0.43$ | $84.50 \pm 0.34$ | $82.94 \pm 0.39$ | $41.81 \pm 0.94$ | $39.40 \pm 0.18$ | $33.26 \pm 0.98$ |
| FedAvg+Fixmatch | $46.71 \pm 2.49$ | $46.67 \pm 0.56$ | $39.95 \pm 1.85$ | $86.46 \pm 0.39$ | $85.42 \pm 0.19$ | $81.07 \pm 0.56$ | $40.40 \pm 0.61$ | $39.66 \pm 1.01$ | $31.99 \pm 0.31$ |
| FedProx+Fixmatch | $47.60 \pm 1.05$ | $43,39 \pm 0.71$ | $41.85 \pm 1.32$ | $86.31 \pm 0.28$ | $85.18 \pm 0.79$ | $83.68 \pm 0.78$ | $41.46 \pm 0.35$ | $40.02 \pm 0.61$ | $32.21 \pm 1.03$ |
| FedMatch | $51.52 \pm 0.50$ | $51.59 \pm 0.32$ | $45.56 \pm 0.91$ | $85.71 \pm 0.21$ | $85.55 \pm 0.09$ | $85.13 \pm 0.15$ | $43.73 \pm 1.15$ | $41.82 \pm 0.23$ | $35.27 \pm 0.35$ |
| **FedDure (Ours)** | $\mathbf{67.69 \pm 0.23}$ | $\mathbf{66.85 \pm 0.65}$ | $\mathbf{57.73 \pm 0.31}$ | $\mathbf{88.69 \pm 0.16}$ | $\mathbf{88.21 \pm 0.07}$ | $\mathbf{86.96 \pm 0.12}$ | $\mathbf{56.36 \pm 0.29}$ | $\mathbf{55.10 \pm 0.25}$ | $\mathbf{46.43 \pm 0.13}$ |

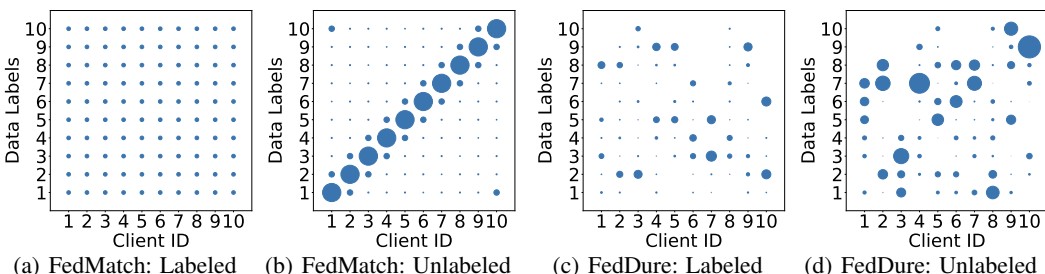

(a) FedMatch: Labeled  (b) FedMatch: Unlabeled  (c) FedDure: Labeled  (d) FedDure: Unlabeled

Figure 2: Comparison of data distribution between FedMatch (Jeong et al., 2021) and our (DIR, DIR) setting: (a) and (b) are labeled and unlabeled data distribution used in FedMatch, respectively; our data distribution in (c) and (d) present external imbalance across clients and internal imbalance between labeled and unlabeled data inside a client.

## 4.1 EXPERIMENTAL SETUP

**Datasets.** We conduct comprehensive experiments on three image classification datasets, including CIFAR-10 (Krizhevsky et al., 2009), Fashion-MNIST (Xiao et al., 2017) and CINIC-10 (Darlow et al., 2018). We provide the simulation of different data heterogeneity for external and internal imbalance below and present more details of these datasets in the Appendix A.1.

**Data Heterogeneity.** We construct three data heterogeneity settings with different data distributions. We denote each setting as $(\mathcal{A}, \mathcal{B})$, where $\mathcal{A}$ and $\mathcal{B}$ are data distribution of labeled and unlabeled data, respectively. The settings are as follows: (1) **(IID, IID)** means both labeled and unlabeled data are IID. By default, we use 5 instances per class to build the labeled dataset for each client. The remaining instances of each class are divided into $K$ clients evenly to build an unlabeled dataset. (2) **(IID, DIR)** means labeled data is the same as (IID, IID), but the unlabeled data is constructed with Dirichlet distribution to simulate data heterogeneity, where each client only contains a subset of classes. (3) **(DIR, DIR)** constructs both labeled and unlabeled data with Dirichlet distribution. It simulates external and internal class imbalance, where the class distributions across clients and within a client are different. Specifically, we allocate 500 labeled data per class to 100 clients using the Dirichlet process; The rest of the instances are also divided into each client with another Dirichlet distribution. Figure 3 compares the data distribution of FedMatch (Batch NonIID) (Jeong et al., 2021) and ours. Our (DIR, DIR) setting presents class imbalance both across clients (external imbalance) and between labeled and unlabeled data within a client (internal imbalance).

**Implementation Detail.** We use the Adam optimizer with momentum $= 0.9$, batch size $= 10$ and learning rates $= 0.0005$ for $\eta_s$, $\eta$ and $\eta_w$. If there is no specified description, our default settings also include local iterations $T = 1$, the selected clients in each round $S = 5$, and the number of clients $K = 100$. For the DIR data configuration, we use a Dirichlet distribution $Dir(\gamma)$ to generate the DIR data for all clients, where $\gamma = 0.5$ for all three datasets. We adopt the ResNet-9 network as the backbone architecture for local models and the coarse-grained regulator, while an MLP is utilized for the fine-grained regulator. More details refer to Appendix A.2.

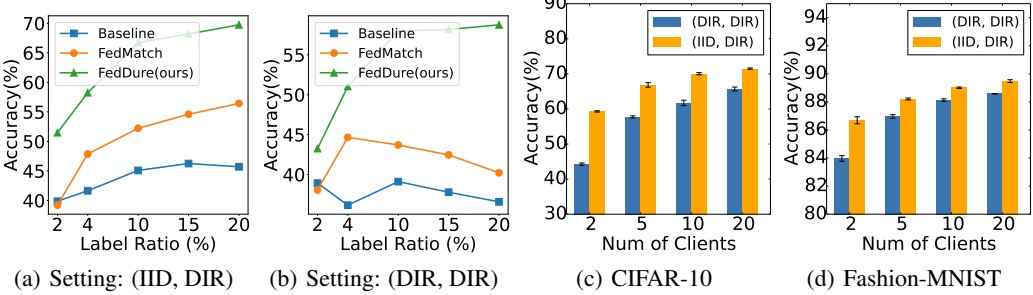

(a) Setting: (IID, DIR)  (b) Setting: (DIR, DIR)  (c) CIFAR-10  (d) Fashion-MNIST

Figure 3: Analysis of the impact of the number of labeled data and the impact of the number of selected clients. (a) and (b) illustrate that FedDure consistently outperforms FedMatch and Baseline (FedAvg-Fixmatch) on different percentages of labeled data. (c) and (d) show that FedDure scales with increasing numbers of selected clients on CIFAR-10 and Fashion-MNIST datasets.

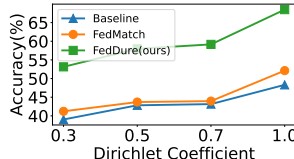

Figure 4: Impact of different Dirichlet coefficients under (DIR, DIR).

Table 2: Quantitative analysis of components of FedDure on CIFAR-10 and Fashion-MNIST datasets.

| Methods | CIFAR-10 | | Fashion-MNIST | |
|---|---|---|---|---|
| | (DIR, DIR) | (IID, DIR) | (DIR, DIR) | (IID, DIR) |
| Baseline | $39.95 \pm 1.85$ | $46.67 \pm 0.56$ | $81.07 \pm 0.56$ | $85.42 \pm 0.19$ |
| *w/o* C-reg | $54.79 \pm 0.59$ | $64.98 \pm 0.58$ | $86.18 \pm 0.13$ | $87.45 \pm 0.23$ |
| *w/o* F-reg | $56.79 \pm 0.63$ | $66.75 \pm 0.23$ | $86.79 \pm 0.17$ | $88.14 \pm 0.12$ |
| FedDure | $\mathbf{57.73 \pm 0.31}$ | $\mathbf{66.85 \pm 0.65}$ | $\mathbf{86.96 \pm 0.12}$ | $\mathbf{88.21 \pm 0.07}$ |

**Baselines.** We compare our FedDure with the following state-of-the-art methods. **FedAvg-SL** (McMahan et al., 2017) and **FedProx-SL** (Li et al., 2020b) denote the supervised algorithms with corresponding FL methods and have fully labeled data in each client. **FedAvg+UDA**, **Fed-Prox+UDA**, **FedAvg+Fixmatch**, and **FedProx+Fixmatch**: a naive combination between semi-supervised method (Sohn et al., 2020; Xie et al., 2020) and FL. Although both labeled and unlabeled data are utilized in these methods, they need to specify a predefined threshold on pseudo labels across decentralized clients. **FedMatch** (Zhang et al., 2021b) adopts inter-consistency loss and disjoint loss for model training, which can reflect state-of-the-art performance in FSSL. Note that, we set equal hyper-parameters for FedDure and other state-of-the-art methods in all experiments.

## 4.2 PERFORMANCE COMPARISON

Table 1 reports the overall results of FedDure and other state-of-the-art methods on the three datasets, where all results are averaged over 3 independent random trails. Our FedDure achieves state-of-the-art FSSL performances on all datasets and data settings. *(IID, IID) setting:* compared with naive combination FSSL methods and FedMatch, our FedDure significantly outperforms them on all three datasets. Specifically, when evaluated on CINIC-10, which is a more difficult dataset and may encounter a larger amount of unlabeled samples, we observe that other methods suffer from the performance bottleneck and are inferior to the evaluation on CIFAR-10 with fewer unlabeled samples. This phenomenon verifies that our FedDure alleviates the influence of mass unlabeled data and prevents performance degradation when the imbalance between labeled and unlabeled data increase rapidly. *(IID, DIR) setting:* our FedDure is slightly affected by weak class mismatch on unlabeled data, but our FedDure makes a rapid performance boost by 16.17% compared to FedMatch on CI-FAR10. Also, competitive performance is achieved compared to the supervised method FedAvg-SL on Fashion-MNIST. *(DIR, DIR) setting:* to simulate the federated semi-supervised scenario in real-world applications, we formulate a severe scenario where labeled data and unlabeled data come from different data distributions and suffer from extreme data imbalance. Under this setting, our FedDure significantly outperforms others by at least 10% on CIFAR-10 and CINIC-10. In particular, we observe that the performance of other approaches degrades sharply and is even inferior to FedAvg* which is only trained on divided labeled data. That is to say, extra unlabeled data might

even have a negative effect on model performance due to the distribution mismatch between labeled and unlabeled data. Therefore, these quantity results demonstrate that our method is well suited for this real-world scenario since the dual regulator effectively and flexibly provides real-time feedback for local model updating.

## 4.3 ABLATION STUDY

**Effectiveness of Components.** To measure the importance of proposed components in our Fed-Dure, we conduct ablation studies with the following variants. (1) baseline: the naive combination of FedAvg and Fixmatch. (2) Ours *w/o* C-reg: this variant removes the C-reg (i.e. $g_d$ in Eqn.12) and updates F-reg with local model. (3) Ours *w/o* F-reg: this variant replaces the dynamic weight (i.e. $g_u$ in Eqn.12) and utilizes the fixed threshold to filter low-confidence pseudo labels. Table 2 shows that adopting the C-reg improves the performance from 54.79% to 57.73% under (DIR, DIR) setting on CIFAR-10. The F-reg can further make a remarkable performance boost under almost all data sets on CIFAR-10 and F-MNIST. These evaluations verify the effectiveness of our components. The local model can flexibly optimize parameters according to the complementary feedback from coarse and fine-grained regulators.

**Number of Label Data per Client.** We evaluate the performance of our method when tuning the proportion of labeled instances in each client in $\{2\%, 4\%, 10\%, 15\%, 20\%\}$. As illustrated in Figure 3(a) and 3(b), we find that our framework gains steady performance improvements with the number of labeled data increases both in two data settings. Interestingly, after the labeling ratio reaches 10%, the performance of the baseline is basically unchanged in (IID, DIR), while we find substantial performance fluctuations for FedMatch and baseline in (DIR, DIR). This phenomenon proves that our regulators can more properly extract valuable knowledge from labeled instances with imbalanced distribution to help local model optimization.

**Number of Selected Clients per Round.** We also investigated the performance impact of the number of selected clients per epoch, which varies in $\{2, 5, 10, 20\}$. As illustrated in Figure 3(c) and 3(d), significant improvements can be achieved by increasing the selected clients. However, there would be a limited impact on performance when the selected clients reach a certain amount. We argue that although the number of the selected clients has a positive correlation with overall performance, our framework can fully explore the underlying knowledge of each client to promote overall performance improvement in the central server. In this case, when there are enough clients, our method has learned comprehensive knowledge such that the performance becomes saturated.

**Impacts of Data Heterogeneity.** As illustrated in Figure 4, our FedDure is the only method that is robust against different levels of internal data imbalance characterized by Dirichlet distribution. FedMatch and baseline (FedAvg-Fixmatch) suffer from the rapid performance degradation in the higher data heterogeneous *(small Dirichlet Coefficient)*. These results demonstrate that our FedDure is more flexible and can alleviate diverse inductive bias across clients when accounting for severe data heterogeneity in real-world applications.

## 5 CONCLUSION

In this paper, we introduce a new federated semi-supervised learning framework with dual regulator, FedDure, to address the challenge of external and internal imbalance of data distribution. Particularly, we propose a coarse-grained regulator to regularize the gradient update in client model training and present a fine-grained regulator to learn an adaptive weighting scheme for unlabeled instances for gradient update. Furthermore, we formulate the learning process in each client as bi-level optimization that optimizes the local model in the client adaptively and dynamically with these two regulators. Theoretically, we show the convergence guarantee of the regulators. Empirically, extensive experiments demonstrate the significance and effectiveness of FedDure. In the future, we consider designing and integrating other client selection strategies for FSSL. Future work also involves extending our method from image classification to more computer vision tasks.

## 6 REPRODUCIBILITY STATEMENT

We provide the datasets, experimental settings, and implementation details in Section 4.1. More details of the experimental setup are provided in Appendix A.1 and A.2. Besides, we summarize our proposed FedDure in Algorithm 1. The source code will be released in the future.

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

# A APPENDIX

## A.1 DATASET

**CIFAR-10** is a dataset containing 60,000 RGB images of 32 x 32 resolutions for image classification tasks. We follow the official guidance and involve 50,000 training images and 10,000 test images. The data set contains 10 categories: "airline", "automobile", "bird", "cat","deer", "dog", "frog", "horse", "ship", "trunk".

**CINIC-10** is a drop-in replacement for CIFAR-10 and is an extension of CIFAR-10 via the addition of downsampled ImageNet images. CINIC-10 contains 270,000 images and is equally split into three subsets: train set, validation set, and test set. In each subset, there are 10 categories like CIFAR-10 and each class involves 9,000 images. In our FedDure training, we apply the training subset for training and allocate them to K clients for uniform or Dirichlet distributions while the global model tests in the test subset.

**Fashion-MINIST** consists of 60,000 training examples and 10,000 test examples. Each example is a 28 x 28 gray-scale image, associated with a label from 10 classes. We implement training in divided training examples and test in test set.

## A.2 IMPLEMENTATION DETAILS

**Network architecture.** Follow the conventional methods Jeong et al. (2021), we employ the official ResNet-9 to local model on all these datasets, i.e, CIFAR-10, CINIC-10 and Fashion-MNIST. Our coarse-grained regulator is a deep copy version of initial local model. Our fine-grained regulator is an MLP architecture, which contains one fully connected layer with 128 filters and follows a Sigmoid function to normalize the output.

**Naive combination methods.** For all reimplement SSL algorithms FixMatch Sohn et al. (2020) or UDA Xie et al. (2020), we fix the confidence threshold 0.85 for all FixMatch and FedMatch methods following their official implementation. To achieve federated semi-supervised methods, the plain FedAvg and FedProx are separately equipped with these SSL methods. For data augmentation, we apply the same strong (RandAugment Cubuk et al. (2020) ) and weak (flip-and-shift) augmentation for unsupervised loss on unlabeled data.

## A.3 ADDITIONAL ABLATION STUDIES

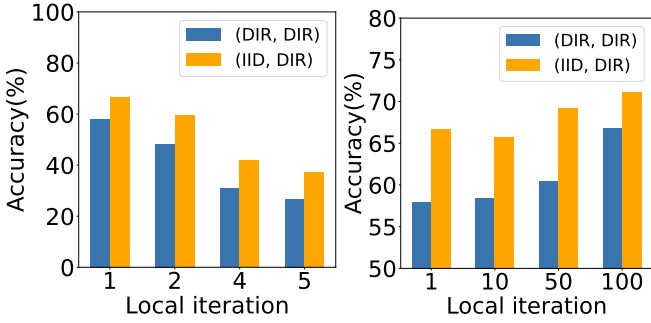

Figure 5: Visualization analysis in terms of different local iterations, where the total number of local iterations or training rounds is fixed.

**Number of Local Iterations.** Finally, to qualitatively analyze the impact of local iterations on performance, we consider two different cases: (1) The total number of local iterations is fixed, where we reduce training rounds with the increase of local iterations to maintain invariant computation cost. For instance, we set local iteration 2, the number of training rounds is 100. As shown in the part of Figure 5, our methods suffer from accuracy degeneration with increasing local iterations. We analyze that while few clients could receive adequate training, many other clients with valuable

information have no chance to be learned, especially with only 40 training rounds. (2) The total number of rounds is fixed, which means we can increase local iterations regardless of computational costs. It can be seen in the right part of Figure 5 that our method achieves steady performance gains by increasing local iterations. These results indicate that enhancing the local model training in each client can promote the overall performance improvements of the central server.

## A.4 DERIVATION OF COARSE-GRAINED REGULATOR

Our C-reg tries to correct the pseudo labels generated by the local model and the local model tries to update its parameters by the feedback from the coarse-grained regulator (C-reg). The cross-entropy difference is applied to quantify the learning effect, it can be derived by gradient-based meta-learning method. We now present the derivation, which theoretically verifies the effectiveness of our coarse-grained regulator. We first formulate the one-step update of the regulator $\boldsymbol{\phi}^{(t)}$ using the sampled soft pseudo label $\hat{\mathbf{y}} \sim f_l(\mathcal{T}_w(\mathbf{u}); \boldsymbol{\theta}_l)$ generated by local model as follows,

$$\boldsymbol{\phi}^{(t+1)}(\boldsymbol{\theta}_l^{(t)}) = \mathbb{E}_{\hat{\mathbf{y}} \sim f_l(\mathcal{T}_w(\mathbf{u}); \boldsymbol{\theta}_l)} \left[ \boldsymbol{\phi}^{(t)} - \eta_s \nabla_{\boldsymbol{\phi}^{(t)}} \mathcal{L}_{ce} \left( \hat{\mathbf{y}}, f_d \left( \mathcal{T}_s(\mathbf{u}); \boldsymbol{\phi}^{(t)} \right) \right) \right].$$

(15)

Then, CE loss on labeled samples $\mathcal{L}_{ce}(\mathbf{y}, f_d(\mathbf{x}; \boldsymbol{\phi}^{(t+1)}(\boldsymbol{\theta}_l^{(t)})))$ is utilized to characterize the quality of pseudo labels from the local model. Since $\boldsymbol{\phi}^{(t+1)}$ has a dependency on $\boldsymbol{\theta}_l^{(t)}$, we can minimize the CE loss to update the local model $\boldsymbol{\theta}_l^{(t)}$ according to the real-time feedback of regulator.

$$\begin{aligned}
\frac{\partial \mathcal{L}}{\partial \theta_l^{(t)}} &= \frac{\partial}{\partial \boldsymbol{\theta}_l^{(t)}} \mathcal{L}_{ce} \left( \mathbf{y}, f_d(\mathbf{x}; \boldsymbol{\phi}^{(t+1)}(\boldsymbol{\theta}_l^{(t)})) \right) \\
&= \frac{\partial \boldsymbol{\phi}^{(t+1)}}{\partial \boldsymbol{\theta}_l^{(t)}} \cdot \frac{\partial}{\partial \boldsymbol{\phi}^{(t+1)}} \mathcal{L}_{ce} \left( \mathbf{y}, f_d(\mathbf{x}; \boldsymbol{\phi}^{(t+1)}) \right)
\end{aligned}$$

(16)

We focus on the first term in Equation 16. Since the $\boldsymbol{\phi}^{(t+1)}$ has no dependency on $\boldsymbol{\theta}_l^{(t)}$ while only $\hat{\boldsymbol{y}}$ depends on $\boldsymbol{\theta}_l^{(t)}$. Note that here $\hat{\boldsymbol{y}}$ is the soft predictions of the local model $f_l$. Therefore, we utilize REINFORCE to achieve,

$$\begin{aligned}
\frac{\partial \boldsymbol{\phi}^{(t+1)}}{\partial \boldsymbol{\theta}_l^{(t)}} &= \frac{\partial}{\partial \boldsymbol{\theta}_l^{(t)}} \mathbb{E}_{\hat{\mathbf{y}} \sim f_l(\mathcal{T}_w(\mathbf{u}); \boldsymbol{\theta}_l)} \left[ \left( \boldsymbol{\phi}^{(t)} - \eta_s \nabla_{\boldsymbol{\phi}^{(t)}} \mathcal{L}_{ce} \left( \hat{\mathbf{y}}, f_d \left( \mathcal{T}_s(\mathbf{u}); \boldsymbol{\phi}^{(t)} \right) \right) \right) \right] \\
&= -\eta_s \cdot \frac{\partial}{\partial \boldsymbol{\theta}_l^{(t)}} \left( \frac{\partial}{\partial \boldsymbol{\phi}^{(t)}} \mathbb{E}_{\hat{\mathbf{y}} \sim f_l(\mathcal{T}_w(\mathbf{u}); \boldsymbol{\theta}_l)} \left[ \mathcal{L}_{ce} \left( \hat{\mathbf{y}}, f_d \left( \mathcal{T}_s(\mathbf{u}); \boldsymbol{\phi}^{(t)} \right) \right) \right] \right) \\
&= -\eta_s \cdot \mathbb{E}_{\hat{\mathbf{y}} \sim f_l(\mathcal{T}_w(\mathbf{u}); \boldsymbol{\theta}_l)} \left[ \frac{\partial}{\partial \boldsymbol{\phi}^{(t)}} \mathcal{L}_{ce} \left( \hat{\mathbf{y}}, f_d \left( \mathcal{T}_s(\mathbf{u}); \boldsymbol{\phi}^{(t)} \right) \right) \cdot \frac{\partial}{\partial \boldsymbol{\theta}_l^{(t)}} \log P \left( \hat{\boldsymbol{y}} | \boldsymbol{u}; \boldsymbol{\theta}_l^{(t)} \right) \right]
\end{aligned}$$

(17)

Then Monte Carlo approximation is applied for the term in Equation 17. We approximate the expected value with the same using $\hat{\boldsymbol{y}}$. Finally, we rewrite the Equation 16 according to the Equation 17 as follows,

$$\frac{\partial \mathcal{L}}{\partial \theta_l^{(t)}} = -\eta_s \frac{\partial}{\partial \boldsymbol{\phi}^{(t+1)}} \mathcal{L}_{ce} \left( \boldsymbol{y}, f_d(\mathbf{x}; \boldsymbol{\phi}^{(t+1)}) \right) \cdot \frac{\partial}{\partial \boldsymbol{\phi}^{(t)}} \mathcal{L}_{ce} \left( \hat{\boldsymbol{y}}, f_d \left( \mathcal{T}_s(\mathbf{u}); \boldsymbol{\phi}^{(t)} \right) \right) \cdot \frac{\partial}{\partial \boldsymbol{\theta}_l^{(t)}} \log P \left( \hat{\boldsymbol{y}} | \boldsymbol{u}; \boldsymbol{\theta}_l^{(t)} \right)$$

(18)

Due to the heavy computation cost, we apply the first-order Taylor expansion to approximate the first two factors. Given that $\boldsymbol{\phi}^{(t+1)} = \boldsymbol{\phi}^{(t)} - \eta_s \frac{\partial}{\partial \boldsymbol{\phi}^{(t)}} \mathcal{L}_{ce} \left( \hat{\mathbf{y}}, f_d \left( \mathcal{T}_s(\mathbf{u}); \boldsymbol{\phi}^{(t)} \right) \right)$, we achieve

$$\begin{aligned}
\mathcal{L}_{ce} \left( \mathbf{y}, f_d(\mathbf{x}; \boldsymbol{\phi}^{(t)}) \right) - \mathcal{L}_{ce} \left( \mathbf{y}, f_d(\mathbf{x}; \boldsymbol{\phi}^{(t+1)}) \right) &= \frac{\partial}{\partial \boldsymbol{\phi}^{(t+1)}} \mathcal{L}_{ce} \left( \mathbf{y}, f_d(\mathbf{x}; \boldsymbol{\phi}^{(t+1)}) \right) \cdot (\boldsymbol{\phi}^{(t)} - \boldsymbol{\phi}^{(t+1)}) \\
&= \eta_s \cdot \frac{\partial}{\partial \boldsymbol{\phi}^{(t+1)}} \mathcal{L}_{ce} \left( \mathbf{y}, f_d(\mathbf{x}; \boldsymbol{\phi}^{(t+1)}) \right) \cdot \frac{\partial}{\partial \boldsymbol{\phi}^{(t)}} \mathcal{L}_{ce} \left( \hat{\mathbf{y}}, f_d \left( \mathcal{T}_s(\mathbf{u}); \boldsymbol{\phi}^{(t)} \right) \right)
\end{aligned}$$

(19)

Now, we rewrite the Equation 18 as follows,

$$
\begin{aligned}
\frac{\partial \mathcal{L}}{\partial \theta_l^{(t)}} &= -\left( \mathcal{L}_{ce}\left( \mathbf{y}, f_d(\mathbf{x}; \boldsymbol{\phi}^{(t)}) \right) - \mathcal{L}_{ce}\left( \mathbf{y}, f_d(\mathbf{x}; \boldsymbol{\phi}^{(t+1)}) \right) \right) \cdot \frac{\partial}{\partial \boldsymbol{\theta}_l^{(t)}} \log P\left( \hat{\mathbf{y}} | \mathbf{u}; \boldsymbol{\theta}_l^{(t)} \right) \\
&= \left( \mathcal{L}_{ce}\left( \mathbf{y}, f_d(\mathbf{x}; \boldsymbol{\phi}^{(t)}) \right) - \mathcal{L}_{ce}\left( \mathbf{y}, f_d(\mathbf{x}; \boldsymbol{\phi}^{(t+1)}) \right) \right) \cdot \frac{\partial}{\partial \boldsymbol{\theta}_l^{(t)}} \mathcal{L}_{ce}\left( \hat{\mathbf{y}}, f_l\left( \mathcal{T}_s(\mathbf{u}); \boldsymbol{\theta}_l^{(t)} \right) \right)
\end{aligned}
\tag{20}
$$

The Equation 10 is equal to Equation 20 and the difference between the coarse-grained regulator characterizes learning difficulty from the perspective of the feedback and learning process of the regulator on labeled samples.

## A.5 CONVERGENCE OF OPTIMIZATION PROCESS

Our FedDure involves a bi-level optimization, so we demonstrate the convergence of these objectives theoretically.

**Theorem 1** *Suppose supervised loss $\mathcal{L}_{ce}(\mathbf{y}, f_d(\boldsymbol{x}; \boldsymbol{\phi}^{(t+1)}(\boldsymbol{\theta}_l^{(t)})))$ is L-Lipschitz and have $\rho$-bounded gradients. The $\mathcal{L}_{ce}\left( \hat{\mathbf{y}}, f_d\left( \mathcal{T}_s(\boldsymbol{u}); \boldsymbol{\phi}^{(t)} \right) \right)$ have $\rho$-bounded gradients and twice differential with Hessian bounded by $\mathcal{B}$. Let the learning rate $\eta_s = \min\{1, \frac{e}{T}\}$ for constant $e > 0$, and $\eta = \min\{\frac{1}{L}, \frac{c}{\sqrt{T}}\}$ for some $c > 0$, such that $\frac{\sqrt{T}}{c} \geq L$. Thus, the optimization of the local model using a coarse-grained regulator can achieve,*

$$
\min_{0 \leq t \leq T} \mathbb{E}[\|\nabla_{\theta_l} \mathcal{L}_{ce}(\mathbf{y}, f_d(\mathbf{x}; \boldsymbol{\phi}^{(t+1)}(\boldsymbol{\theta}_l^{(t)})))\|_2^2] \leq \mathcal{O}(\frac{c}{\sqrt{T}}),
\tag{21}
$$

The CE loss $\mathcal{L}_{ce}(\mathbf{y}, f_d(\mathbf{x}; \boldsymbol{\phi}^{(t+1)}(\boldsymbol{\theta}_l^{(t)})))$ is minimized to update the local model $\boldsymbol{\theta}_l^{(t)}$. Let $\hat{L}(\boldsymbol{\phi}^{(t+1)}; \boldsymbol{\theta}^{(t)}) = \mathcal{L}_{ce}(\mathbf{y}, f_d(\mathbf{x}; \boldsymbol{\phi}^{(t+1)}(\boldsymbol{\theta}_l^{(t)})))$, so the update of $\boldsymbol{\theta}_l$ in each step as follows,

$$
\boldsymbol{\theta}^{(t+1)} = \boldsymbol{\theta}^{(t)} - \eta \nabla_\theta \hat{L}(\boldsymbol{\phi}^{(t+1)}; \boldsymbol{\theta}^{(t)})
\tag{22}
$$

In coarse-grained optimization process, the updating for $\boldsymbol{\theta}^{(t)}$ to $\boldsymbol{\theta}^{(t+1)}$ is,

$$
\begin{aligned}
&\hat{L}(\boldsymbol{\phi}^{(t+2)}; \boldsymbol{\theta}^{(t+1)}) - \hat{L}(\boldsymbol{\phi}^{(t+1)}; \boldsymbol{\theta}^{(t)}) \\
&= \{\hat{L}(\boldsymbol{\phi}^{(t+2)}; \boldsymbol{\theta}^{(t+1)}) - \hat{L}(\boldsymbol{\phi}^{(t+1)}; \boldsymbol{\theta}^{(t+1)})\} + \{\hat{L}(\boldsymbol{\phi}^{(t+1)}; \boldsymbol{\theta}^{(t+1)}) - \hat{L}(\boldsymbol{\phi}^{(t+1)}; \boldsymbol{\theta}^{(t)})\}
\end{aligned}
\tag{23}
$$

For the first term, since we have $\boldsymbol{\phi}^{(t+1)} = \boldsymbol{\phi}^{(t)} - \eta_s \nabla_{\boldsymbol{\phi}^{(t)}} \mathbb{E}_{\mathbf{u}} \mathcal{L}_{ce}\left( \hat{\mathbf{y}}, f_d\left( \mathcal{T}_s(\mathbf{u}); \boldsymbol{\phi}^{(t)} \right) \right)$, let $\mathcal{V}^{coarse}\left( \boldsymbol{\phi}^{(t)} \right) = \mathbb{E}_{\mathbf{u}} \mathcal{L}_{ce}\left( \hat{\mathbf{y}}, f_d\left( \mathcal{T}_s(\mathbf{u}); \boldsymbol{\phi}^{(t)} \right) \right)$, we achieve,

$$
\begin{aligned}
&\hat{L}(\boldsymbol{\phi}^{(t+2)}; \boldsymbol{\theta}^{(t+1)}) - \hat{L}(\boldsymbol{\phi}^{(t+1)}; \boldsymbol{\theta}^{(t+1)}) \\
&\leq \langle \nabla_{\phi^{(t+1)}} \mathcal{V}^{coarse}(\boldsymbol{\phi}^{(t+1)}), -\eta_s \nabla_{\phi^{(t+1)}} \mathcal{V}^{coarse}(\boldsymbol{\phi}^{(t+1)}) \rangle + \frac{L}{2} \|\boldsymbol{\phi}^{(t+2)} - \boldsymbol{\phi}^{(t+1)}\|_2^2 \\
&= \langle \nabla_{\phi^{(t+1)}} \mathcal{V}^{coarse}(\boldsymbol{\phi}^{(t+1)}), -\eta_s \nabla_{\phi^{(t+1)}} \mathcal{V}^{coarse}(\boldsymbol{\phi}^{(t+1)}) \rangle + \frac{L}{2} \| -\eta_s \nabla_{\phi^{(t+1)}} \mathcal{V}^{coarse}(\boldsymbol{\phi}^{(t+1)})\|_2^2 \\
&= \left( -\eta_s + \frac{\eta_s^2 L}{2} \right) \|\nabla_{\phi^{(t+1)}} \mathcal{V}^{coarse}(\boldsymbol{\phi}^{(t+1)})\|_2^2
\end{aligned}
\tag{24}
$$

Since $\nabla_{\phi^{(t+1)}} \mathcal{V}^{coarse}(\boldsymbol{\phi}^{(t+1)}) \leq \rho$, so Equation 24 satisfies,

$$
\hat{L}(\boldsymbol{\phi}^{(t+2)}; \boldsymbol{\theta}^{(t+1)}) - \hat{L}(\boldsymbol{\phi}^{(t+1)}; \boldsymbol{\theta}^{(t+1)}) \leq \left( -\eta_s + \frac{\eta_s^2 L}{2} \right) \rho^2
\tag{25}
$$

For the second term, we have,

$$
\hat{L}(\boldsymbol{\phi}^{(t+1)};\boldsymbol{\theta}^{(t+1)}) - \hat{L}(\boldsymbol{\phi}^{(t+1)};\boldsymbol{\theta}^{(t)})
$$

$$
\leq \langle \nabla_{\theta^{(t)}} \hat{L}(\boldsymbol{\phi}^{(t+1)};\boldsymbol{\theta}^{(t)}), \boldsymbol{\theta}^{(t+1)} - \boldsymbol{\theta}^{(t)} \rangle + \frac{L}{2} \|\boldsymbol{\theta}^{(t+1)} - \boldsymbol{\theta}^{(t)}\|_2^2
$$

$$
= \langle \nabla_{\theta^{(t)}} \hat{L}(\boldsymbol{\phi}^{(t+1)};\boldsymbol{\theta}^{(t)}), -\eta \nabla_{\theta^{(t)}} \hat{L}(\boldsymbol{\phi}^{(t+1)};\boldsymbol{\theta}^{(t)}) \rangle + \frac{L}{2} \| -\eta \nabla_{\theta^{(t)}} \hat{L}(\boldsymbol{\phi}^{(t+1)};\boldsymbol{\theta}^{(t)})\|_2^2 \tag{26}
$$

$$
= \left( -\eta + \frac{\eta^2 L}{2} \right) \|\nabla_{\theta^{(t)}} \hat{L}(\boldsymbol{\phi}^{(t+1)};\boldsymbol{\theta}^{(t)})\|_2^2
$$

Combining these two terms we have,

$$
\hat{L}(\boldsymbol{\phi}^{(t+2)};\boldsymbol{\theta}^{(t+1)}) - \hat{L}(\boldsymbol{\phi}^{(t+1)};\boldsymbol{\theta}^{(t)}) \leq \left( -\eta_s + \frac{\eta_s{}^2 L}{2} \right)\rho^2 + \left( -\eta + \frac{\eta^2 L}{2} \right) \|\nabla_{\theta^{(t)}} \hat{L}(\boldsymbol{\phi}^{(t+1)};\boldsymbol{\theta}^{(t)})\|_2^2 \tag{27}
$$

Summing up all iterations, we can obtain,

$$
\hat{L}(\boldsymbol{\phi}^{(T+2)};\boldsymbol{\theta}^{(T+1)}) - \hat{L}(\boldsymbol{\phi}^{(2)};\boldsymbol{\theta}^{(1)}) \leq \sum_{t=1}^{T}\left( -\eta_s + \frac{\eta_s{}^2 L}{2} \right)\rho^2 + \sum_{t=1}^{T}\left( -\eta + \frac{\eta^2 L}{2} \right) \|\nabla_{\theta^{(t)}} \hat{L}(\boldsymbol{\phi}^{(t+1)};\boldsymbol{\theta}^{(t)})\|_2^2
$$

$$
\sum_{t=1}^{T}\left( \eta - \frac{\eta^2 L}{2} \right) \|\nabla_{\theta^{(t)}} \hat{L}(\boldsymbol{\phi}^{(t+1)};\boldsymbol{\theta}^{(t)})\|_2^2 \leq \sum_{t=1}^{T}\left( -\eta_s + \frac{\eta_s{}^2 L}{2} \right)\rho^2 - \hat{L}(\boldsymbol{\phi}^{(T+2)};\boldsymbol{\theta}^{(T+1)}) + \hat{L}(\boldsymbol{\phi}^{(2)};\boldsymbol{\theta}^{(1)})
$$

$$
\leq \sum_{t=1}^{T}\left( -\eta_s + \frac{\eta_s{}^2 L}{2} \right)\rho^2 + \hat{L}(\boldsymbol{\phi}^{(2)};\boldsymbol{\theta}^{(1)})
$$

(28)

Furthermore, we deduce that,

$$
\min_{t} \mathbb{E}[\|\nabla_{\theta^{(t)}} \hat{L}(\boldsymbol{\phi}^{(t+1)};\boldsymbol{\theta}^{(t)})\|_2^2] \leq \frac{1}{\sum_{t=1}^{T}\left( \eta - \frac{\eta^2 L}{2} \right)} \left( \sum_{t=1}^{T}\left( -\eta_s + \frac{\eta_s{}^2 L}{2} \right)\rho^2 + \hat{L}(\boldsymbol{\phi}^{(2)};\boldsymbol{\theta}^{(1)}) \right)
$$

$$
\leq \frac{1}{\sum_{t=1}^{T}\eta} \left( \sum_{t=1}^{T}\left( -2\eta_s + \eta_s{}^2 L \right)\rho^2 + 2\hat{L}(\boldsymbol{\phi}^{(2)};\boldsymbol{\theta}^{(1)}) \right)
$$

$$
= \frac{\sum_{t=1}^{T}\left( -2\eta_s + \eta_s{}^2 L \right)\rho^2}{\sum_{t=1}^{T}\eta} + \frac{2\hat{L}(\boldsymbol{\phi}^{(2)};\boldsymbol{\theta}^{(1)})}{\sum_{t=1}^{T}\eta}
$$

$$
= \frac{\left( -2\eta_s + \eta_s{}^2 L \right)\rho^2}{\eta} + \frac{2\hat{L}(\boldsymbol{\phi}^{(2)};\boldsymbol{\theta}^{(1)})}{T\eta}
$$

$$
= \left( -2\eta_s + \eta_s L \right)\rho^2 \max\{L, \frac{\sqrt{T}}{c}\} + \frac{2\hat{L}(\boldsymbol{\phi}^{(2)};\boldsymbol{\theta}^{(1)})}{T} \max\{L, \frac{\sqrt{T}}{c}\}
$$

$$
= \left( L - 2 \right)\rho^2 \min\{1, \frac{e}{T}\} \max\{L, \frac{\sqrt{T}}{c}\} + \frac{2\hat{L}(\boldsymbol{\phi}^{(2)};\boldsymbol{\theta}^{(1)})}{T} \max\{L, \frac{\sqrt{T}}{c}\}
$$

$$
\leq \frac{e(L-2)\rho^2}{c\sqrt{T}} + \frac{2\hat{L}(\boldsymbol{\phi}^{(2)};\boldsymbol{\theta}^{(1)})}{c\sqrt{T}} = \mathcal{O}\left( \frac{c}{\sqrt{T}} \right)
$$

(29)

The proof has been completed.

Let the $\boldsymbol{\phi}$ is optimized with the loss $L^{fine}(\boldsymbol{\phi}^{(t)};\boldsymbol{w}^{(t+1)}) = \mathcal{H}(\boldsymbol{w}^{(t+1)};\boldsymbol{\phi}^{(t)})\mathcal{L}_{ce}\left( \hat{\mathbf{y}}, f_d\left( \mathcal{T}_s(\mathbf{u});\boldsymbol{\phi}^{(t)} \right) \right)$, where $\mathcal{H}(\boldsymbol{w}^{(t+1)};\boldsymbol{\phi}^{(t)}) = f_w\left( f_d\left( \mathcal{T}_s(\mathbf{u});\boldsymbol{\phi}_l^{(t)} \right);\boldsymbol{w}^{(t+1)} \right)$ indicates the adaptive weight for samples $\boldsymbol{u}$.

**Theorem 2** *Suppose supervised and unsupervised loss functions are Lipschitz-smooth with constant L and have $\rho$-bounded gradient. The $\mathcal{H}(\cdot)$ is differential with a $\epsilon$-bounded gradient and twice differential with its Hessian bounded by $\mathcal{B}$. Let learning rate $\eta_s$ satisfies $\eta_s = \min\{1, \frac{k}{T}\}$ for constant*

$k > 0$, *such that* $\frac{k}{T} < 1$. $\eta_w = \min\{\frac{1}{L}, \frac{c}{\sqrt{T}}\}$ *for constant* $c > 0$ *such that* $\frac{\sqrt{T}}{c} \geq L$. *The method can achieve*

$$\lim_{t \to \infty} \mathbb{E}[\|\nabla_{\phi^{(t)}} L^{fine}(\phi^t; w^{(t+1)})\|_2^2] = 0. \tag{30}$$

The proof. The optimization from $w^{(t)}$ to $w^{(t+1)}$ is,

$$w^{(t+1)} = w^{(t)} - \eta_w \nabla_{w^{(t)}} \underbrace{\mathcal{L}_{ce}\left(\mathbf{y}, f_w\left(f_d(\mathbf{x}; \phi^{(t)}); w^{(t)}\right)\right)}_{:=\mathcal{L}^s(\boldsymbol{x}; \phi^{(t)}, w^{(t)})},$$

(31)

We know the updating of $\phi$ is,

$$\phi^{(t+1)} = \phi^{(t)} - \eta_s \underbrace{\mathcal{H}(w^{(t+1)}; \phi^{(t)}) \nabla_\phi \left. \mathcal{L}_{ce}\left(\hat{\mathbf{y}}, f_d\left(\mathcal{T}_s(\mathbf{u}); \phi^{(t)}\right)\right)\right|_{\phi^{(t)}}}_{:=\nabla_\phi \mathcal{L}^u(\boldsymbol{u}; \phi^{(t)}, w^{(t+1)})}.$$

(32)

Observe that,

$$L^{fine}(\phi^{(t+1)}; w^{(t+2)}) - L^{fine}(\phi^t; w^{(t+1)})$$
$$= \{L^{fine}(\phi^{(t+1)}; w^{(t+2)}) - L^{fine}(\phi^{(t+1)}; w^{(t+1)})\} + \{L^{fine}(\phi^{(t+1)}; w^{(t+1)}) - L^{fine}(\phi^t; w^{(t+1)})\}. \tag{33}$$

For the first term, we have

$$L^{fine}(\phi^{(t+1)}; w^{(t+2)}) - L^{fine}(\phi^{(t+1)}; w^{(t+1)})$$
$$= \left(\mathcal{H}(w^{(t+2)}; \phi^{(t+1)}) - \mathcal{H}(w^{(t+1)}; \phi^{(t+1)})\right) \mathcal{L}_{ce}\left(\hat{\mathbf{y}}, f_d\left(\mathcal{T}_s(\mathbf{u}); \phi^{(t+1)}\right)\right). \tag{34}$$

In the first factor, we achieve,

$$\mathcal{H}(w^{(t+2)}; \phi^{(t+1)}) - \mathcal{H}(w^{(t+1)}; \phi^{(t+1)})$$
$$\leq \langle \nabla_{w^{(t+1)}} \mathcal{H}(w^{(t+1)}; \phi^{(t+1)}), w^{(t+2)} - w^{(t+1)} \rangle + \frac{L}{2} \|w^{(t+2)} - w^{(t+1)}\|_2^2$$
$$= \langle \nabla_{w^{(t+1)}} \mathcal{H}(w^{(t+1)}; \phi^{(t+1)}), -\eta_w \nabla_{w^{(t+1)}} \mathcal{L}^s(\boldsymbol{x}; \phi^{(t+1)}, w^{(t+1)}) \rangle + \frac{L}{2} \| - \eta_w \nabla_{w^{(t)}} \mathcal{L}^s(\boldsymbol{x}; \phi^{(t+1)}, w^{(t+1)}) \|_2^2$$
$$= -\eta_w \cdot \nabla_{w^{(t+1)}} \mathcal{H}(w^{(t+1)}; \phi^{(t+1)}) \cdot \nabla_{w^{(t+1)}} \mathcal{L}^s(\boldsymbol{x}; \phi^{(t+1)}, w^{(t+1)}) + \frac{\eta_w^2 L}{2} \|\nabla_{w^{(t+1)}} \mathcal{L}^s(\boldsymbol{x}; \phi^{(t+1)}, w^{(t+1)})\|_2^2. \tag{35}$$

For the second term, we have

$$L^{fine}(\phi^{(t+1)}; w^{(t+1)}) - L^{fine}(\phi^{(t)}; w^{(t+1)})$$
$$\leq \langle \nabla_{\phi^{(t)}} L^{fine}(\phi^{(t)}; w^{(t+1)}), \phi^{(t+1)} - \phi^{(t)} \rangle + \frac{L}{2} \|\phi^{(t+1)} - \phi^{(t)}\|_2^2$$
$$= \langle \nabla_{\phi^{(t)}} L^{fine}(\phi^{(t)}; w^{(t+1)}), -\eta_s \nabla_\phi \mathcal{L}^u(\boldsymbol{u}; \phi^{(t)}, w^{(t+1)}) \rangle + \frac{L}{2} \| - \eta_s \nabla_\phi \mathcal{L}^u(\boldsymbol{u}; \phi^{(t)}, w^{(t+1)}) \|_2^2$$
$$= -\eta_s \|\nabla_{\phi^{(t)}} \mathcal{L}^u(\boldsymbol{u}; \phi^{(t)}, w^{(t+1)})\|_2^2 + \frac{\eta_s^2 L}{2} \|\nabla_{\phi^{(t)}} \mathcal{L}^u(\boldsymbol{u}; \phi^{(t)}, w^{(t+1)})\|_2^2. \tag{36}$$

Combining these two terms we have,

$$L^{fine}(\boldsymbol{\phi}^{(t+1)}; \boldsymbol{w}^{(t+2)}) - L^{fine}(\boldsymbol{\phi}^{(t)}; \boldsymbol{w}^{(t+1)})$$

$$\leq -\eta_s \|\nabla_{\phi^{(t)}} \mathcal{L}^u(\boldsymbol{u}; \boldsymbol{\phi}^{(t)}, \boldsymbol{w}^{(t+1)})\|_2^2 + \frac{\eta_s^2 L}{2} \|\nabla_{\phi^{(t)}} \mathcal{L}^u(\boldsymbol{u}; \boldsymbol{\phi}^{(t)}, \boldsymbol{w}^{(t+1)})\|_2^2$$

$$+ \mathcal{L}_{ce}\left(\hat{\mathbf{y}}, f_d\left(\mathcal{T}_s(\mathbf{u}); \boldsymbol{\phi}^{(t+1)}\right)\right) \cdot \{-\eta_w \cdot \nabla_{w^{(t+1)}} \mathcal{H}(\boldsymbol{w}^{(t+1)}; \boldsymbol{\phi}^{(t+1)}) \cdot \nabla_{\boldsymbol{w}^{(t+1)}} \mathcal{L}^s(\boldsymbol{x}; \boldsymbol{\phi}^{(t+1)}, \boldsymbol{w}^{(t+1)})$$

$$+ \frac{\eta_w^2 L}{2} \|\nabla_{\boldsymbol{w}^{(t+1)}} \mathcal{L}^s(\boldsymbol{x}; \boldsymbol{\phi}^{(t+1)}, \boldsymbol{w}^{(t+1)})\|_2^2\}. \tag{37}$$

Rearranging the inequality, we obtain,

$$\eta_w \cdot \nabla_{w^{(t+1)}} \mathcal{H}(\boldsymbol{w}^{(t+1)}; \boldsymbol{\phi}^{(t+1)}) \cdot \nabla_{\boldsymbol{w}^{(t+1)}} \mathcal{L}^s(\boldsymbol{x}; \boldsymbol{\phi}^{(t+1)}, \boldsymbol{w}^{(t+1)}) \cdot \mathcal{L}_{ce}\left(\hat{\mathbf{y}}, f_d\left(\mathcal{T}_s(\mathbf{u}); \boldsymbol{\phi}^{(t+1)}\right)\right)$$

$$+ \eta_s \|\nabla_{\phi^{(t)}} \mathcal{L}^u(\boldsymbol{u}; \boldsymbol{\phi}^{(t)}, \boldsymbol{w}^{(t+1)})\|_2^2 \leq \frac{\eta_s^2 L}{2} \|\nabla_{\phi^{(t)}} \mathcal{L}^u(\boldsymbol{u}; \boldsymbol{\phi}^{(t)}, \boldsymbol{w}^{(t+1)})\|_2^2$$

$$+ \frac{\eta_w^2 L}{2} \|\nabla_{\boldsymbol{w}^{(t+1)}} \mathcal{L}^s(\boldsymbol{x}; \boldsymbol{\phi}^{(t+1)}, \boldsymbol{w}^{(t+1)})\|_2^2 \cdot \mathcal{L}_{ce}\left(\hat{\mathbf{y}}, f_d\left(\mathcal{T}_s(\mathbf{u}); \boldsymbol{\phi}^{(t+1)}\right)\right)$$

$$+ L^{fine}(\boldsymbol{\phi}^{(t)}; \boldsymbol{w}^{(t+1)}) - L^{fine}(\boldsymbol{\phi}^{(t+1)}; \boldsymbol{w}^{(t+2)}) \tag{38}$$

Summing up the inequalities in $T$ iterations on both sides, we achieve

$$\sum_{t=1}^{T} \eta_w \cdot \nabla_{w^{(t+1)}} \mathcal{H}(\boldsymbol{w}^{(t+1)}; \boldsymbol{\phi}^{(t+1)}) \cdot \nabla_{\boldsymbol{w}^{(t+1)}} \mathcal{L}^s(\boldsymbol{x}; \boldsymbol{\phi}^{(t+1)}, \boldsymbol{w}^{(t+1)}) \cdot \mathcal{L}_{ce}\left(\hat{\mathbf{y}}, f_d\left(\mathcal{T}_s(\mathbf{u}); \boldsymbol{\phi}^{(t+1)}\right)\right)$$

$$+ \sum_{t=1}^{T} \eta_s \|\nabla_{\phi^{(t)}} \mathcal{L}^u(\boldsymbol{u}; \boldsymbol{\phi}^{(t)}, \boldsymbol{w}^{(t+1)})\|_2^2 \leq \frac{\eta_s^2 L}{2} \sum_{t=1}^{T} \|\nabla_{\phi^{(t)}} \mathcal{L}^u(\boldsymbol{u}; \boldsymbol{\phi}^{(t)}, \boldsymbol{w}^{(t+1)})\|_2^2$$

$$+ \frac{\eta_w^2 L}{2} \sum_{t=1}^{T} \|\nabla_{\boldsymbol{w}^{(t+1)}} \mathcal{L}^s(\boldsymbol{x}; \boldsymbol{\phi}^{(t+1)}, \boldsymbol{w}^{(t+1)})\|_2^2 \cdot \mathcal{L}_{ce}\left(\hat{\mathbf{y}}, f_d\left(\mathcal{T}_s(\mathbf{u}); \boldsymbol{\phi}^{(t+1)}\right)\right)$$

$$+ L^{fine}(\boldsymbol{\phi}^{(1)}; \boldsymbol{w}^{(2)}) - L^{fine}(\boldsymbol{\phi}^{(T+1)}; \boldsymbol{w}^{(T+2)})$$

$$\leq \frac{\eta_s^2 L}{2} \sum_{t=1}^{T} \|\nabla_{\phi^{(t)}} \mathcal{L}^u(\boldsymbol{u}; \boldsymbol{\phi}^{(t)}, \boldsymbol{w}^{(t+1)})\|_2^2$$

$$+ \frac{\eta_w^2 L}{2} \sum_{t=1}^{T} \|\nabla_{\boldsymbol{w}^{(t+1)}} \mathcal{L}^s(\boldsymbol{x}; \boldsymbol{\phi}^{(t+1)}, \boldsymbol{w}^{(t+1)})\|_2^2 \cdot \mathcal{L}_{ce}\left(\hat{\mathbf{y}}, f_d\left(\mathcal{T}_s(\mathbf{u}); \boldsymbol{\phi}^{(t+1)}\right)\right)$$

$$+ L^{fine}(\boldsymbol{\phi}^{(1)}; \boldsymbol{w}^{(2)})$$

$$\leq \frac{\eta_s^2 L T \rho^2}{2} + \frac{\eta_w^2 L T \rho^2}{2} \sum_{t=1}^{T} \mathcal{L}_{ce}\left(\hat{\mathbf{y}}, f_d\left(\mathcal{T}_s(\mathbf{u}); \boldsymbol{\phi}^{(t+1)}\right)\right) + L^{fine}(\boldsymbol{\phi}^{(1)}; \boldsymbol{w}^{(2)}) \tag{39}$$

When $T \to \infty$, we can obtain,

$$\sum_{t=1}^{T} \eta_w \cdot \nabla_{w^{(t+1)}} \mathcal{H}(\boldsymbol{w}^{(t+1)}; \boldsymbol{\phi}^{(t+1)}) \cdot \nabla_{\boldsymbol{w}^{(t+1)}} \mathcal{L}^s(\boldsymbol{x}; \boldsymbol{\phi}^{(t+1)}, \boldsymbol{w}^{(t+1)}) \cdot \mathcal{L}_{ce}\left(\hat{\mathbf{y}}, f_d\left(\mathcal{T}_s(\mathbf{u}); \boldsymbol{\phi}^{(t+1)}\right)\right)$$

$$+ \sum_{t=1}^{T} \eta_s \|\nabla_{\phi^{(t)}} \mathcal{L}^u(\boldsymbol{u}; \boldsymbol{\phi}^{(t)}, \boldsymbol{w}^{(t+1)})\|_2^2$$

$$\leq \lim_{T \to \infty} \sum_{t=1}^{T} \eta_w \epsilon \rho \cdot \mathcal{L}_{ce}\left(\hat{\mathbf{y}}, f_d\left(\mathcal{T}_s(\mathbf{u}); \boldsymbol{\phi}^{(t+1)}\right)\right) + \lim_{T \to \infty} \sum_{t=1}^{T} \eta_s \|\nabla_{\phi^{(t)}} \mathcal{L}^u(\boldsymbol{u}; \boldsymbol{\phi}^{(t)}, \boldsymbol{w}^{(t+1)})\|_2^2$$

$$\leq \lim_{T \to \infty} \frac{\eta_s^2 L T \rho^2}{2} + \lim_{T \to \infty} \frac{\eta_w^2 L T \rho^2}{2} \sum_{t=1}^{T} \mathcal{L}_{ce}\left(\hat{\mathbf{y}}, f_d\left(\mathcal{T}_s(\mathbf{u}); \boldsymbol{\phi}^{(t+1)}\right)\right) + L^{fine}(\boldsymbol{\phi}^{(1)}; \boldsymbol{w}^{(2)}) \leq \infty, \tag{40}$$

since $\mathcal{L}_{ce}\left(\hat{\mathbf{y}}, f_d\left(\mathcal{T}_s(\mathbf{u}); \boldsymbol{\phi}^{(t+1)}\right)\right)$ is bounded for a limited number of unlabeled samples, supervised loss function have $\rho$-bounded gradient and the $\mathcal{H}()$ is differential with a $\epsilon$-bounded gradient. Therefore we deduce that

$$\lim_{T \to \infty} \sum_{t=1}^{T} \eta_s \|\nabla_{\phi^{(t)}} \mathcal{L}^u(\boldsymbol{u}; \boldsymbol{\phi}^{(t)}, \boldsymbol{w}^{(t+1)})\|_2^2 < \infty. \tag{41}$$

According to the Lemma A.5 Mairal (2013), to prove $\lim_{t \to \infty} \mathbb{E}[\|\nabla_{\phi^{(t)}} \mathcal{L}^u(\boldsymbol{u}; \boldsymbol{\phi}^{(t)}, \boldsymbol{w}^{(t+1)})\|_2^2] = 0$, we should prove

$$\left| \mathbb{E}[\|\nabla_{\phi^{(t)}} \mathcal{L}^u(\boldsymbol{u}; \boldsymbol{\phi}^{(t+1)}, \boldsymbol{w}^{(t+2)})\|_2^2] - \mathbb{E}[\|\nabla_{\phi^{(t)}} \mathcal{L}^u(\boldsymbol{u}; \boldsymbol{\phi}^{(t)}, \boldsymbol{w}^{(t+1)})\|_2^2] \right| \le Q\eta_s, \tag{42}$$

where $Q$ represents constant. Consider the Equation 41, we have

$$\begin{aligned}
&\left| \mathbb{E}[\|\nabla_{\phi^{(t+1)}} \mathcal{L}^u(\boldsymbol{u}; \boldsymbol{\phi}^{(t+1)}, \boldsymbol{w}^{(t+2)})\|_2^2] - \mathbb{E}[\|\nabla_{\phi^{(t)}} \mathcal{L}^u(\boldsymbol{u}; \boldsymbol{\phi}^{(t)}, \boldsymbol{w}^{(t+1)})\|_2^2] \right| \\
&\le \mathbb{E}[(\|\nabla_{\phi^{(t+1)}} \mathcal{L}^u(\boldsymbol{u}; \boldsymbol{\phi}^{(t+1)}, \boldsymbol{w}^{(t+2)})\|_2 + \|\nabla_{\phi^{(t)}} \mathcal{L}^u(\boldsymbol{u}; \boldsymbol{\phi}^{(t)}, \boldsymbol{w}^{(t+1)})\|_2) \\
&\quad \cdot (\|\nabla_{\phi^{(t+1)}} \mathcal{L}^u(\boldsymbol{u}; \boldsymbol{\phi}^{(t+1)}, \boldsymbol{w}^{(t+2)})\|_2 - \|\nabla_{\phi^{(t)}} \mathcal{L}^u(\boldsymbol{u}; \boldsymbol{\phi}^{(t)}, \boldsymbol{w}^{(t+1)})\|_2)] \\
&\le \mathbb{E}[(\|\nabla_{\phi^{(t+1)}} \mathcal{L}^u(\boldsymbol{u}; \boldsymbol{\phi}^{(t+1)}, \boldsymbol{w}^{(t+2)}) + \nabla_{\phi^{(t)}} \mathcal{L}^u(\boldsymbol{u}; \boldsymbol{\phi}^{(t)}, \boldsymbol{w}^{(t+1)})\|_2) \\
&\quad \cdot (\|\nabla_{\phi^{(t+1)}} \mathcal{L}^u(\boldsymbol{u}; \boldsymbol{\phi}^{(t+1)}, \boldsymbol{w}^{(t+2)}) - \nabla_{\phi^{(t)}} \mathcal{L}^u(\boldsymbol{u}; \boldsymbol{\phi}^{(t)}, \boldsymbol{w}^{(t+1)})\|_2)] \\
&\le 2L\rho\eta_s\eta_w \sqrt{\mathbb{E}[\|\nabla_{\phi^{(t)}} \mathcal{L}^u(\boldsymbol{u}; \boldsymbol{\phi}^{(t)}, \boldsymbol{w}^{(t+1)})\|_2^2] + \mathbb{E}[\|\nabla_{w^{(t)}} \mathcal{L}^s(\boldsymbol{x}; \boldsymbol{\phi}^{(t)}, \boldsymbol{w}^{(t)})\|]} \\
&\le 2\sqrt{2}L\rho^2\eta_s\eta_w
\end{aligned} \tag{43}$$

Thus, it has proved. Since $L^{fine}(\boldsymbol{\phi}^t; \boldsymbol{w}^{(t+1)}) = \mathcal{L}^u(\boldsymbol{u}; \boldsymbol{\phi}^{(t)}, \boldsymbol{w}^{(t+1)})$, therefore, the proof has completed.

