# OpenReview forum: "Federated Semi-supervised Learning with Dual Regulator"
_ICLR.cc/2023/Conference — Submitted to ICLR 2023_

### Official Review · Reviewer_JdwN · 2022-10-24

**Confidence:** 3
**Correctness:** 2
**Technical Novelty And Significance:** 2
**Empirical Novelty And Significance:** 2
**Recommendation:** 5

**Clarity, Quality, Novelty And Reproducibility:**

While the idea and the concept of the C-reg and F-reg are novel, unfortunately, the paper is not clearly written and probably has issues. More specifically, the core section 3 has conflicting information; for example, in eq.3 the authors provide one update rule for the parameters of C-reg, however, in eq. 6 they provide another. Which one is used in practice and why? Furthermore, there seems to also be a disconnect between what is described in Algorithm 1, lines 16-20 and what is described in the “Meta-Process” paragraph in section 3.3; in the latter it is mentioned that C-reg is updated first, with F-reg second and then C-reg is updated once more (and I guess all of this before updating the actual local model) whereas in the algorithm the steps are a) optimize F-reg, b) optimise C-reg for fixed F-reg, c) optimise the local model. Besides that, there are critical information missing with respect to, e.g., $f_w()$; how is that defined and what is its output? From the boldface notation at eq.8 it seems that its output is the same as the dimensionality of the gradient vector? Given these concerns, I do not believe that this work is easily reproducible.

Besides that, I also found the more detailed derivations in the appendix unclear. For example in Appendix A.4, the authors mention that they use REINFORCE and Monte-Carlo, however there is no expectation / randomness as, from what I understand, $p(\hat{y})$ is a delta peak (obtained from the argmax in eq. 1 of the main paper). Furthermore, it seems that eq.17 is missing a negative sign and from eq.19 to the next one, the difference of cross-entropies is is changed to the difference of entropies. For this to happen, you would need to replace the empirical distribution over the label (y) with the model distribution $f_s()$ and I do not follow how this happens from eq. 19. Furthermore, the proof for Theorem 1 starts by mentioning that the local model is optimised using the CE loss involving the parameters of the coarse-regulator. As the local model is updated using eq. 11 with gradients of various losses, I am not sure what the guarantee in theorem 1 is about and whether it is at all useful (as it seems to be about a different procedure than the one done at Alg. 1). I have similar concerns for Theorem 2 where the proof seems to ignore the fact that $\phi$, $w$ are affected by $\theta$. In addition, there are some peculiar statements near the end of page 19 that the $L_{CE}()$ “converges”. What do the authors mean by that? Furthermore, how is eq. 39 obtained? By taking the limit, shouldn’t you have something like $\lim_{T->\inf} \sum_{t=1}^T L_{CE}(\phi^{t+1})$, instead of some constants times the $L_{CE}$ using the $\phi$ parameters at an arbitrary time $t+1$?


**Strength And Weaknesses:**

Strengths:
- Good performance on the tasks considered
- An ablation study was performed

Weaknesses:
- Paper is not clearly written and can be hard to follow
- Theory seems to be disconnected from the actual method
- FedDure has a lot of moving parts
- There are no error bars on the performance metrics, so it can be hard to determine the significance of the results

**Summary Of The Paper:**

This work proposes FedDure, a method for semi-supervised learning (SSL) in federated learning. FedDure employs two “regulators” in order to improve performance in SSL scenarios. The first regulator is the C(oarse) reg(ulator); its job is to regularise the local model training by taking into account the pseudo-labels the local model assigns to unlabelled data. It is a network (initialised to be the network received from the server) which is trained on the pseudo-labelled dataset, where the pseudo labels are obtained from network received from the server. The authors perform a single step of training for the C-reg and then use the difference of the predictive entropies on the actual labelled data as a weight for the gradient of the local model on the cross-entropy loss of the pseudo-labelled data.  The second regulator is the F(ine) reg(ulator); its job is to learn an instance specific weight for the loss that models the impact each unlabelled datapoint should have in training. The training signal for this weight is the cross-entropy loss of the C-reg on the labelled data. The authors alternate between optimising the F-reg, optimising the C-reg for a fixed F-reg and then updating the actual local model using gradient contributions from the labelled data and gradient contributions from the unlabelled data that depend on the C-reg and F-reg. The overall architecture is evaluated on cifar10, cinic10 and fashion mnist, using various split settings that lead to data heterogeneity in both the labelled and unlabelled data.


**Summary Of The Review:**

While the concept of the F-reg and C-reg are novel and the experiments seem to show improvements upon the baselines (although without error-bars so their significance is a bit unclear), I cannot recommend acceptance given the aforementioned concerns.

---

> ### Author Response · Authors · 2022-11-13
> **Reply to Reviewer JdwN (1/2)**
>
> Thank you for the thoughtful review. You raise several salient points that help us significantly improve this work. We address your questions below.
>
> **Q1**: Which update rules is used for the parameters of C-reg, eq.3 or eq.6?
>
> **A1**: Thank you for raising the question. The update rule of C-reg is eq.6. Eq.3 represents a conceptual approximate of $\phi^*$ based on $\theta_l$ , which establishes the meta-learning process between these two hyper-parameters. Eq.6 is the actual formula that we use to update C-reg. We have revised Section 3.2 and 3.3 to make them clearer. Eq. 6 is now Eq. 4 in the manuscript after revision.
>
> **Q2**: About the disconnect between discription in Algorithm 1 and what is described in the "Meta-Process" in section 3.3.
>
> **A2**: Thank you for the constructive comment. We would like to clarify that the order of updates follows Algorithm 1. The one-step update of C-reg is an intermediate variable for the update of F-reg. We have revised Section 3.3 to prevent potential misunderstanding.
>
> **Q3**: How is $f_w()$ defined and what is its output?
>
> **A3**: $f_w$  is the F-reg we have defined in Section 3.2 and it is a fully connected layer with 128  filters and a Sigmoid function (provided in Appendix A.2). We added a footnote when introducing $f_w$ to make it clearer.
> The input $f_w$ is the normalized logits of a batch of unlabeled samples and the output is the corresponding scalar weights for every unlabeled labels, and is not the same as the dimensionality of the gradient vector.
>
> **Q4**: About the use of REINFORCE.
>
> **A4**: Thank you for raising the question. We have followed your suggestion and revised the derivation of C-reg in Appendix A.4. The local model samples $\hat{y}$ from the soft predictions of the local model on the unlabeled instances $\pmb{u}$, which can be denoted as $\hat{y} \sim f_l(\mathcal{T}_w( \textbf{u}); \pmb{\theta}_l)$. When using soft pseudo labels, we use REINFORCE algorithm in expectation.
>
> **Q5**: About the comments on C-reg derivation.
>
> **A5**: Thank you for pointing this out! We have followed your advice and added a negative sign in eq.17. $f_s$ is a typo; we actually mean $f_d$ , which represents C-reg. We have revised the manuscript and the Appendix following your advice.
>
> **Q6**: As the local model is updated using eq. 11 with gradients of various losses, I am not sure what the guarantee in theorem 1 is about and whether it is at all useful (as it seems to be about a different procedure than the one done at Alg. 1.
>
> **A6**: We have clarified the procedure in the previous questions and revised the manuscript. Hope it resolves the doubt of Reviewer JdwN on the procedure of optimization. Theorem 1 provides the convergence guarantee of $\pmb{g}_d$. Besides, Eq. 11 (Now Eq. 12 after revision) contains three losses: 1) loss to update local model with C-reg $\pmb{g}_d$; 2) loss to update local model with F-reg  $\pmb{g}_u$. It is a consistency regularization loss that is well studied in prior works like [1], while we incorporate a unique weight $\pmb{m}_i$ for each unlabeled sample when calculating the loss. ; 3) supervised loss on labeled data $\pmb{g}_s$. Thus, we primarily focus on providing the convergence guarantee of $\pmb{g}_d$ in Theorem 1.
>
> [1] Laine, Samuli, and Timo Aila. "Temporal ensembling for semi-supervised learning." ICLR 2017.

---

> > ### Author Response · Authors · 2022-11-13
> > **Reply to Reviewer JdwN (2/2)**
> >
> > **Q7**: The proof in Theorem 2 seems to ignore the fact that $\phi,w$ are affected by $\theta$.
> >
> > **A7**: We would like to clarify that $\phi$ and $w$ are not directly affected by $\theta$ in Theorem 2. $\mathcal{H} (\pmb{w}^{(t+1)};\pmb{\phi}^{(t)})$ in Eq. 4 is the weights on unlabeled samples; The optimization of C-reg regards the $\theta$ of local model $f_l$ as fixed parameters.
> >
> > **Q8**: The peculiar statements near the end of page 19.
> >
> > **A8**: Thank you for pointing this out. We mean that
> > $\mathcal{L}_{ce}\left(\hat{\textbf{y}}, f_s\left(\mathcal{T}_s(\textbf{u}); \pmb{\phi}^{(t+1)}\right)\right) \leq C$ for the CE loss on a limited number of unlabeled samples is bounded. We have revised it in the manuscript.
> >
> > **Q9**: How is eq.39 obtained?
> >
> > **A9**: Thank you for raising the question. Eq. 39 (Now Eq. 40 in the manuscript) is obtained from Eq. 38 (Now Eq. 39 in the manuscript) as the assumption is that the supervised loss function has $\rho$-bounded gradient and the $\mathcal{H}()$ is differential with a $\epsilon$-bounded gradient. We have further clarified it in the manuscript to prevent potential misunderstanding.
> >
> > **Q10**: By taking the limit, shouldn't you have something like  $\lim_{T\rightarrow \infty} \sum_{t=1}^T L_{CE}(\phi^{(t+1)})$?
> >
> > Thank you for your punctilious comment. We have followed your suggestion to revise it and the inequality in the equation still also holds.
> >
> > **Q11**: There are no error bars on the performance metrics, so it can be hard to determine the significance of the results.
> >
> > **A11**: Thanks for the concern. We have followed your suggestion and added error bars for our method and other comparison algorithms in Table 1, Table 2, Figure 3(c), and Figure 3(d).
> >
> > **Q12**: I do not believe that this work is easily reproducible.
> >
> > **A12**: Thank you for raising the concern. We have revised Sections 3.2 and 3.3 to make our manuscript clearer and have provided other details in Section 4.1 and Appendixes. The codes will be released after our internal processes.

---

> ### Author Response · Authors · 2022-11-17
> **Look forward to your feedback!**
>
> Dear Reviewer JdwN,
>
> We sincerely thank you for your time and efforts in reviewing our paper, and appreciate your detailed and constructive comments. We have carefully revised the manuscript by incorporating your suggestions, and provided additional experimental results. Since the rebuttal will be ended only by this Friday (November 18), we would like to kindly remind you to check our responses and the revised manuscript. We hope it can address your concerns and look forward to your feedback.
>
> Best regards,
>
> Authors of Paper1558

---

> ### Comment · Reviewer_JdwN · 2022-11-18
> **Response to rebuttal**
>
> I would like to thank the authors for responding to my review. Based on their response, I have decided to increase my score to a 5. I still cannot fully recommend acceptance due to clarity / not well motivated assumptions. For example, the authors argue that $L_{CE}() \leq C$ for a limited number of unlabelled samples, however, the cross entropy loss is not bounded from above so, even if you have a single unlabelled datapoint, there is a possibility of $L_{CE}() > C$.

---

> ### Author Response · Authors · 2022-11-19
> **Further Reply to Reviewer JdwN**
>
> We would like to thank you for raising the score, which affirms that the quality of the paper is improved after we revised it following reviewer's constructive comments. The assumption of the equation is that the cross entropy loss is bounded on a finite unlabelled dataset $\mathcal{D}^u$, i.e. $\exists C < \infty, \forall u \in \mathcal{D}^u, \mathcal{L}_{ce}\left(\hat{\textbf{y}}, f_s\left(\mathcal{T}_s(\textbf{u}); \pmb{\phi}^{(t+1)}\right)\right) \leq C$. We will continue to polish the paper and improve its clarity. Thank you.

---

### Official Review · Reviewer_yeCB · 2022-10-26

**Confidence:** 2
**Clarity, Quality, Novelty And Reproducibility:** See the above for the evaluation.
**Correctness:** 3
**Technical Novelty And Significance:** 3
**Empirical Novelty And Significance:** 3
**Recommendation:** 6

**Strength And Weaknesses:**

### Strength
1. The proposed method is technically sound.
2. Experiments show that the proposed method significantly outperforms existing methods for different scenarios.
3. The paper is overall well presented.

### Weakness
1. The proposed dual-regulator is now very new; we can see similar techniques in many learning problems, e.g., semi-supervised learning, few-shot learning, learning with noisy labels. etc. With that being said, I still appreciate the value of introducing and tailoring these techniques to the federated learning scenario.
2. Many federated learning works evaluate different data types, images, texts, etc, which would definitively gives more comprehensive evaluation of the proposed method. I do not see the proposed method has an assumption for the data type and thus would be better if such experiments can be conducted.
3.  There are a few typos and inconsistency of using capitalization in the paper, which should be corrected in the future: sometimes all the words are capitalized for the first letter for the title and sometimes not. The language use is not very professional and should be improved.

**Summary Of The Paper:**

This paper studies federated semi-supervised learning and propose a new method with a dual-regulator which can handle the heterogeneity of labeled data and unlabeled data from each client. The main contribution is the proposed dual-regulator where the first regulator updates local models  in an meta-learning based way by accessing the updating effect, the second regulator assigns weights to unlabeled instances. Experiments shows that the proposed method significantly outperforms existing methods for different data distribution scenarios.

**Summary Of The Review:**

I am not an expert in this field,  my assessment is based on educational guess. Overall this paper looks good: it studies a problem which was overlooked by previous method, proposes a method which is technically sound, and produce significantly better performance.

---

> ### Author Response · Authors · 2022-11-13
> **Reply to Reviewer yeCB**
>
> Thank you for your time and effort in reviewing our paper, and giving us some valuable suggestions. We respond to individual comments below:
>
> **Q1**: The proposed dual-regulator is now very new; we can see similar techniques in many learning problems, e.g., semi-supervised learning, few-shot learning, learning with noisy labels. etc. With that being said, I still appreciate the value of introducing and tailoring these techniques to the federated learning scenario.
>
> **A1**: Thank you for the comment. We are inspired by some of the existing works, but simply adopting those works into FL does not work well. We really appreciate the reviewer for acknowledging the contributions to our work.
>
> **Q2**: Many federated learning works evaluate different data types, images, texts, etc, which would definitively gives more comprehensive evaluation of the proposed method. I do not see the proposed method has an assumption for the data type and thus would be better if such experiments can be conducted.
>
> **A2**: Thank you for the constructive comment. We would like to clarify that we primarily focus on image datasets because the pseudo-label generation of images and texts is fundamentally different. It involves weak and strong augmentation of image data, which may not be directly applied to text datasets. Many pioneering works [1] [2] also mainly focus on image datasets. It is exciting to extend our dual-regulator concept to the text dataset and we will investigate it in our future works. We have revised the manuscript to make our assumption clearer.
>
> [1] Sohn, Kihyuk, et al. "Fixmatch: Simplifying semi-supervised learning with consistency and confidence." NeurIPS 2020.
>
> [2] Jeong, Wonyong, et al. "Federated semi-supervised learning with inter-client consistency & disjoint learning." ICLR 2021.
>
> **Q3**: There are a few typos and inconsistency of using capitalization in the paper, which should be corrected in the future: sometimes all the words are capitalized for the first letter for the title and sometimes not. The language use is not very professional and should be improved.
>
> **A3**: Thank you for raising the issues. We have followed your advice and revised the manuscript.

---

> ### Author Response · Authors · 2022-11-17
> **Look forward to your feedback!**
>
> Dear Reviewer yeCB,
>
> Could you please go over our responses and the revised version of paper since the rebuttal is ending soon? We have carefully revised the manuscript by incorporating your suggestions. We sincerely hope it can address your concerns and look forward to your feedback.
>
> Best regards,
>
> Authors of Paper1558

---

### Official Review · Reviewer_DPn5 · 2022-10-26

**Confidence:** 3
**Correctness:** 3
**Technical Novelty And Significance:** 3
**Empirical Novelty And Significance:** 3
**Recommendation:** 6

**Clarity, Quality, Novelty And Reproducibility:**

The method is novel. Details and motivations of the method needs to be clarified further.

**Strength And Weaknesses:**

Strengths:
- This work proposes a novel method that handles a realistic scenario that haven't been studied before.
- This work provides formal convergence guarantee.
- The empirical results look promising.

Weaknesses:
- It is unclear what is the motivation for having both coarse-grained and fine-grained regulator. From figure 1 and Section 3, it's hard to follow what is the purpose of having each of them.
- Is there any theoretical guarantee on how the proposed method handles imbalanced data distribution between labeled and unlabeled data?
- I'm curious how the method performs in cross-device setting where the number of clients is large and each client has low probability to be selected at each round for scalability. The results would be more convincing if such results are provided.

Nit:
- In Equation (2), should it be argmin over $\phi$ rather than argmax?

**Summary Of The Paper:**

In this work, the authors propose a new method for federated semi-supervised learning relying on a coarse grained regulator and fine grained regulator. This work handles scenarios where the labeled data and unlabeled data share different distribution. Empirical results show that the proposed method is able to outperform prior baselines.

**Summary Of The Review:**

This work tackles an interesting case where the labeled data and unlabeled data comes from different distribution. This problem is conceptually realistic and interesting. The proposed method also shows strong empirical performance.

---

> ### Author Response · Authors · 2022-11-13
> **Reply to Reviewer DPn5**
>
> We appreciate your thoughtful comments and constructive suggestions, which will improve the presentation of our work. We respond to your comments below:
>
> **Q1**: It is unclear what is the motivation for having both coarse-grained and fine-grained regulator. From figure 1 and Section 3, it's hard to follow what is the purpose of having each of them.
>
> **A1**: Thank you for raising the concern. These two regulators tackle two challenges that are not well addressed by prior work: 1) Local training on unlabeled data could be substantially hindered by incorrect pseudo labels, especially in the early period of training rounds. 2) The data distribution across clients is heterogeneous (external and internal imbalance). Using a fixed threshold would result in an extreme imbalance of filtered pseudo labels in clients.
>
> Specifically, the coarse-grained regulator (C-reg) and the fine-grained regulator (F-reg) adaptively regularize the model training in each client from two complementary perspectives: C-reg dynamically regulates the importance of local training on the unlabeled data by reflecting the overall learning effect on labeled data; F-reg regulates the performance contribution of each unlabeled sample.
>
> **Q2**: Is there any theoretical guarantee on how the proposed method handles imbalanced data distribution between labeled and unlabeled data?
>
> **A2**: Thank you for raising the question. We have provided theoretical proof to show that the proposed method is under convergence guarantee in local training and empirically demonstrate that this work handles imbalanced data distribution well. It could be a new work to further quantify and analyze different data distributions between labeled and unlabeled data. For example, the pioneering work of FL, FedAvg [1], proposes a new method; many following works provide a theoretical analysis of the method under different data distributions [2]. We will adopt your suggestion and further investigate it in future work.
>
> [1] McMahan, Brendan, et al. "Communication-efficient learning of deep networks from decentralized data." AISTATS 2017.
>
> [2] Li, Xiang, et al. "On the convergence of fedavg on non-iid data." ICLR 2020.
>
> **Q3**: How the method performs in cross-device setting where the number of clients is large and each client has low probability to be selected at each round for scalability. The results would be more convincing if such results are provided.
>
> **A3**: Thanks for the constructive comment. We would like to clarify that the majority of experiments are conducted under the cross-device setting, where 5 clients are selected out of 100 clients in each training round. We also would like to invite Reviewer DPn5 to revisit Figure 3 (c) and (d), where we provide more results of different numbers of selected clients ({2, 5, 10, 20} out of 100 clients) in each training round. Please let us know the additional experiment setting you are interested in, we will provide results for that and incorporate them into the manuscript.

---

> ### Author Response · Authors · 2022-11-17
> **Look forward to your feedback!**
>
> Dear Reviewer DPn5,
>
> We sincerely thank you for your precious time and efforts in reviewing our paper, and your insightful and constructive comments. We have carefully revised the manuscript and and highlighted the changes in the new version. Since the rebuttal time is ending soon, we would like to kindly remind you to check our response and the revised version of our paper. We hope it can address your concerns and look forward to your feedback.
>
> Thanks,
>
> Authors of Paper1558

---

> ### Comment · Reviewer_DPn5 · 2022-12-03
> **Re: Response**
>
> Thank you for your responses. That resolves most of my questions. I agree that further studying heterogeneous data distribution between labeled and unlabeled data theoretically would be important and potentially strengthen the paper.

---

### Author Response · Authors · 2022-11-13
**General Response**

We thank the reviewers for their thoughtful feedback! We are encouraged they find the problem is overlooked by the previous method (Reviewer yeCB and DPn5) and conceptually realistic and interesting (Reviewer DPn5). Also, the proposed method is novel (Reviewer DPn5) and technically sound (Reviewer yeCB). Besides, we are glad that all reviewers agree that the empirical results are promising, and have significantly better performance than the existing methods. We are also encouraged that they find the paper is well presented and overall looks good (Reviewer yeCB). We answer reviewers’ specific comments in our responses and have incorporated all feedback in the revised manuscript.

---

### Decision · Program_Chairs · 2023-01-20

**Decision:**

Reject

**Justification For Why Not Higher Score:**

there is not enough motivation and discussion about their results.

**Justification For Why Not Lower Score:**

n/a

**Metareview: Summary, Strengths And Weaknesses:**

This paper proposes a new learning framework for semi-supervised data in FL environment. In particular, the authors target the situation where label data as well as unlabeled data possessed by each client are heterogeneous.

Despite the excellent performance compared to the baselines shown in the experiments, I believe that this paper is not self-contained and  is not ready for publication because the sections are not coherent and well motivated. For examples,
- There is no mention of what kind of heterogeneity is considered in the problem setting.
- In Section 3 motivation of each component of the proposed method is not provided at all. Only methods are simply described. Therefore, it is very difficult for readers to understand why each component is proposed. In a similar context, it is difficult to understand why the performance is so much better than the existing methods even in the iid environment.
- Section 4 also simply provides the theorems (and proofs in the appendix). No discussion is provided; it is impossible to grasp the meaning of each theorem and how it can be compared with existing studies etc.

Reviewers also positively evaluated the performance improvement, but there was an opinion that it was difficult to grasp the content, and there was no reviewer who strongly supported the acceptance of this paper through discussion, so I propose a rejection.